# Estimating Dietary Intake from Grocery Shopping Data—A Comparative Validation of Relevant Indicators in Switzerland

**DOI:** 10.3390/nu14010159

**Published:** 2021-12-29

**Authors:** Jing Wu, Klaus Fuchs, Jie Lian, Mirella Lindsay Haldimann, Tanja Schneider, Simon Mayer, Jaewook Byun, Roland Gassmann, Christine Brombach, Elgar Fleisch

**Affiliations:** 1Interaction- and Communication-based Systems, Institute of Computer Science, University of St. Gallen, 9000 St. Gallen, Switzerland; jing.wu@unisg.ch (J.W.); jie.lian@unisg.ch (J.L.); simon.mayer@unisg.ch (S.M.); 2ETH AI Center, ETH Zurich, Swiss Federal Institute of Technology in Zurich, 8092 Zurich, Switzerland; 3D-One Solutions AG, 8003 Zurich, Switzerland; mirella.haldimann@d-one.ai; 4Technology Studies, School of Humanities and Social Sciences, University of St. Gallen, 9000 St. Gallen, Switzerland; 5Department of Software, Sejong University, Seoul 05006, Korea; jwbyun@sejong.ac.kr; 6Institute of Food and Beverage Innovation, Zurich University of Applied Sciences, 8820 Waedenswil, Switzerland; garo@zhaw.ch (R.G.); broc@zhaw.ch (C.B.); 7Information Management, Department of Management, Technology, and Economics, Swiss Federal Institute of Technology in Zurich, 8092 Zurich, Switzerland; efleisch@ethz.ch; 8Technology Management, Institute of Technology Management, University of St. Gallen, 9000 St. Gallen, Switzerland

**Keywords:** food shopping quality indicators, FSA-NPS DI, dietary intake, diet monitoring, digital receipts

## Abstract

In light of the globally increasing prevalence of diet-related chronic diseases, new scalable and non-invasive dietary monitoring techniques are urgently needed. Automatically collected digital receipts from loyalty cards hereby promise to serve as an objective and automatically traceable digital marker for individual food choice behavior and do not require users to manually log individual meal items. With the introduction of the General Data Privacy Regulation in the European Union, millions of consumers gained the right to access their shopping data in a machine-readable form, representing a historic chance to leverage shopping data for scalable monitoring of food choices. Multiple quantitative indicators for evaluating the nutritional quality of food shopping have been suggested, but so far, no comparison has validated the potential of these alternative indicators within a comparative setting. This manuscript thus represents the first study to compare the calibration capacity and to validate the discrimination potential of previously suggested food shopping quality indicators for the nutritional quality of shopped groceries, including the Food Standards Agency Nutrient Profiling System Dietary Index (FSA-NPS DI), Grocery Purchase Quality Index-2016 (GPQI), Healthy Eating Index-2015 (HEI-2015), Healthy Trolley Index (HETI) and Healthy Purchase Index (HPI), checking if any of them performs differently from the others. The hypothesis is that some food shopping quality indicators outperform the others in calibrating and discriminating individual actual dietary intake. To assess the indicators’ potentials, 89 eligible participants completed a validated food frequency questionnaire (FFQ) and donated their digital receipts from the loyalty card programs of the two leading Swiss grocery retailers, which represent 70% of the national grocery market. Compared to *absolute* food and nutrient intake, correlations between density-based *relative* food and nutrient intake and food shopping data are stronger. The FSA-NPS DI has the best calibration and discrimination performance in classifying participants’ consumption of nutrients and food groups, and seems to be a superior indicator to estimate nutritional quality of a user’s diet based on digital receipts from grocery shopping in Switzerland.

## 1. Introduction

The globally increasing prevalence of diet-related chronic diseases, including obesity, diabetes and certain types of cancers, represents a growing burden for affected patients and health-care systems alike [1,2,3,4]. Due to societal trends, such as urbanization and the transformation of food systems toward more processed and convenience food items, dietary patterns around the world show an increase in consumed (added) sugar, sodium, saturated fats, and calorific energy [5]. These increases elevate the risks of diet-related chronic diseases [6,7,8]. Besides the uptake of food items of low nutritional quality, the global demand for meat, fish, and exotic fruits throughout the year takes a significant toll on the planetary ecosystem as well [1]. To counter the global trend toward unhealthy and unsustainable food choices, novel automatic and scalable monitoring tools are needed that have the potential to help deal with the burden of unhealthy dietary patterns, monitor and eventually improve food choices [9].

### 1.1. Conventional Diet Monitoring Approaches

To analyze an individual’s dietary pattern, dietitians and researchers typically rely on self-reported and laborious dietary assessment approaches, such as seven-day weighed food diaries or records, 24 h recalls and food frequency questionnaires (FFQs) [10,11,12,13]. Although individual dietary data collected in these ways have an accepted accuracy, the required manual transcription is prone to recall biases and high attrition rates, particularly when the data collection is conducted over longer periods of time [9,14,15,16]. The European Food Safety Authority (EFSA) promoted the use of software-based applications for FFQs, 24 h recalls and food diaries to reduce the labor intensity of data collection [17]. Even so, the user attrition and adoption in such self-tracking applications remain challenging. For instance, manual diet logging apps (e.g., *MyFitnessPal*) are only actively used by 8% of smartphone users [18]. In addition, collecting data on individual dietary behavior can be expensive, especially within a validated setup, e.g., conducting FFQs under supervision or assessing sodium excretion from 24 h urine collection or measuring micro-nutrient content from blood samples.

Given that contemporary diet monitoring techniques suffer from short-lived retention, memory bias, low acceptance and expensive costs, it is hard to conduct longitudinal studies with a strong statistical power or even population-wide diet monitoring via self-reporting dietary monitoring methods only. To allow for more inclusive and continuous food choice monitoring, new scalable and automated food choice monitoring tools to substitute or complement contemporary dietary monitoring approaches are needed.

### 1.2. Digital Receipts

With the proliferation of digital payments and loyalty cards, digital receipts from grocery shopping can serve as a novel, scalable, automatically and continuously self-updating monitoring tool for food choices [19,20]. Digital receipts can be seen as machine-readable, electronic substitutes for their contemporary paper-based printed counterparts. They are expected to be adopted around the globe over the next decade, as they promise significant advantages with regard to environmental footprint [21] and mitigating tax evasion [22,23], offering superior advantages and transparency for consumers [24].

Food shopping comprises a major part of people’s shopping in supermarkets. Assuming that consumers eat a majority of the groceries which they buy with their loyalty cards, there exists the possibility to infer individual eating behavior from the household shopping records. Compared to conventional diet monitoring tools which focus on collecting individual dietary intake data, digital receipts do not require the laborious active logging of every single meal, and all historic purchase records are available instantly. In addition to data on purchased quantities and types of selected food items, information about expenditures and corresponding timestamps is available. Compared to contemporary food monitoring approaches, the multi-dimensional digital receipt dataset thus provides new possibilities to explore further aspects of participants’ food choice behavior. These facets can include their favorite stores and shopping habits, their price sensitivity, their preferences for specific brands, categories and flavors, or even their desire to purchase seasonably, regional or sustainable products, all potentially relevant avenues for diet-related interventions.

Although representing partial data in a household context, digital receipts can serve as a fully automated diet monitoring proxy for estimating individual food intake. An important distinction between digital receipts and the aforementioned contemporary diet monitoring tools is that digital receipts usually represent the shopping behavior of a household rather than of an individual. This is because households, e.g., a family or a shared flat, tend to share groceries and loyalty cards. Assessing household level food shopping data to infer insights on individual food intake behavior results in lower but still acceptable accuracy, compared to individual dietary monitoring tools (e.g., FFQs, food diaries, bio-samples, such as blood or urine sampling). The challenges are manifold. First, food shopping data are incomplete, and do not consider process factors such as food preparation, out-of-home consumption, delayed consumption and food waste [25,26]. Still, the purchases from supermarkets usually represent the vast majority of dietary intake. For example, in the case of sodium, which is the most often consumed via eating packaged food products shopped from supermarkets, an estimation of 80% of its intake originates from supermarket purchases [27]. Second, the conversion from household-level food shopping data to individual diet behavior is challenging. Nevertheless, previous studies have repeatedly demonstrated that applying statistical methods on the partial food shopping data allows the inference of absolute dietary intake as well as relative distributions of individual food choices [28,29]. Food purchases captured by receipts correlate with and can predict individual dietary patterns [20,28,30,31,32,33], demonstrating moderate to strong agreement [20,31,32,34,35,36,37]. The dietary calorific intake of a person correlates with the amount of shopped energy-dense food products [34,35,37], or that obesity could be detected via analyzing food shopping patterns [31]. More concretely, by assessing their relative distribution in terms of weight, calories or expenditures, household-level food shopping data can be converted to individual level dietary intake estimates. These weight-based, expenditure-weighted or calorie-weighted approaches thus implicitly assume that each person in the household consumes similar proportions of the purchased food categories. While this assumption might be justified for single and smaller households, it might lack validity for very large households. Still, given the literature in the field, digital receipts can be assessed as a scalable, non-invasive proxy for individual dietary intake behavior.

In the past, research in this field has been strongly limited by the restricted access to digital receipt data. This is because considerable efforts are involved in collecting product data and sample sizes are often small. A key aspect of monitoring food choices via digital receipts is collecting a user’s electronic shopping history via loyalty cards, which previously was often done manually via collecting printed paper receipts, or taking stock of home inventory of shopped food items [19,28,29,38,39]. Because of the introduction of the General Data Protection Regulation (GDPR) [40], millions of customers in the European Union recently gained the right to access their digital receipts from loyalty cards. Consequentially, researchers may obtain these data with their study subjects’ consents. Therefore, product shopping, including food shopping recorded on loyalty cards, is now retrievable from data processors (i.e., retailers and loyalty card providers). By itself, a digital receipt does not contain any nutritional details of the purchased food products. To this end, food product composition databases, which were mandated by the regulation on mandatory declaration of food information for food items sold online (EU)1169 [41], allow the data fusion of digital receipts and nutritional information on the shopped food products.

To objectively determine whether a shopping history record of a certain user indicates a healthy behavior, validated quantitative indicators are needed to offer a normative, reliable and interpretable basis for comparison between users, households, regions and retailers.

### 1.3. Proposed Food Shopping Quality Indicators

Multiple quantitative food shopping indicators have been suggested to evaluate the nutritional quality of food and beverage shopping records, including the Grocery Purchase Quality Index-2016 (GPQI) [28] (designed and validated within the United States (US)), the Healthy Trolley Index (HETI) [29] (designed and validated within Australia) and the Healthy Purchase Index (HPI) [42] (designed and validated within France). These three suggested indicators all assign the observed food items in the shopping data into different food group categories, thereby allowing to estimate how balanced each respective user’s shopping history is. Taking their respective local dietary guidelines as the golden standards, these indicators might not necessarily be directly ’transferable’ to all regions, but are valid in their respective geographies. In addition, the expenditure share of the respective food groups was used in the calculation of these three indices in this study’s digital receipt dataset from Switzerland to guarantee the compliance with respect to each indicator’s corresponding index guidelines. In absence of food composition data, price-based indicators might be an interim solution as suggested by the HETI, HPI and GPQI indices. However, if available, shopped quantities in grams or milliliters or energy in kilocalories (kcal) or kilojoules (kJ) would give a more accurate representation of the dietary impact from the evaluated shopping records than price-based estimations. Additionally, traditional diet indices can be used to evaluate food shopping behavior. The Healthy Eating Index-2010 (HEI-2010) scores derived from food shopping data showed moderate agreement and minimal bias with HEI-2010 scores from 24 h recalls [43]. Thus, using food shopping data to calculate Healthy Eating Index-2015 (HEI-2015) [44], the latest version of HEI, might be a feasible way of assessing nutritional quality from digital receipts. Finally, the Food Standards Agency Nutrient Profiling System Dietary Index (FSA-NPS DI) [45] can also be utilized to assess food shopping quality. Instead of assigning food to different food groups and evaluating how compliant the baskets are to a certain dietary guideline, FSA-NPS DI takes all food items into account, weighs them based on the calorific contribution and focuses on the overall nutritional quality of the entire basket, in this case, series of shopping baskets over the study’s observation period.

Despite the existence of multiple shopping indices, a published assessment validating the calibration and discrimination potential of these indicators in a comparative environment does not yet exist. The calibration capacity of a model, or an indicator in this case, can be reflected by how agreeable the prediction and actual outcomes are. The discrimination capacity evaluates how well a model can perform in separating cases with and without certain outcomes [46,47]. The main objective of this study is to compare the calibration and discrimination ability of the aforementioned quantitative shopping indices, namely FSA-NPS DI, GPQI, HEI-2015, HETI and HPI. Digital receipts that were automatically captured from loyalty cards in Switzerland are used as the data for the calculation of the five indicators. The results derived from the validated FFQs were taken as the objective measurement and were what the food shopping quality indicators were used to calibrate and discriminate. The hypothesis is that some of them perform significantly better than the others in a general situation. The results should equip researchers, practitioners, and policymakers with the insights required to select one among the existing indices. The conclusions might be relevant for researchers and practitioners who work on designing novel food shopping quality indicators, monitoring systems or interventions in this domain.

## 2. Materials and Methods

This manuscript describes the first study comparing the calibration capacity and validating the discrimination potential of multiple previously suggested food shopping quality indicators for the nutritional quality of shopped groceries, including FSA-NPS DI, GPQI, HEI-2015, HETI and HPI. To be more specific, the calibration capacity indicates how closely the food shopping quality indicators calculated from digital receipts and the dietary intake reflected by the FFQ results are correlated. The discrimination capacity shows how well the food shopping quality indicators can distinguish people with different levels of dietary intake. In the following, the digital receipt integration, food composition database and study design are introduced.

### 2.1. Digital Receipt Integration

The digital receipt infrastructure was implemented in Switzerland due to the availability of digital receipts from the loyalty card systems of the two leading supermarket chains. To support the comparative analysis of the suggested quantitative food shopping quality indicators, a technical setup was designed and implemented to allow the collection of receipts from users who consented to participate in the study. The study was deployed on the Bitsaboutme (BAM) online platform (see https://bitsabout.me/, accessed on 15 December 2021). BAM is a GDPR-compliant data marketplace service located in Switzerland, which allows users to request their own personal user data from data controllers and store their data in an encrypted data vault. Just as data sources such as social media or messaging services process personal data, financial transactions from bank accounts and digital receipts are also considered as personal data by the GDPR. Hence, users of the BAM service are able to retrieve their digital receipts from the two leading Swiss loyalty card providers, namely Migros Cumulus and Coop Supercard. In this regard, Switzerland can be considered an ideal region to validate quantitative food shopping quality indicators, as just these two leading Swiss grocery chains, i.e., Migros and Coop, represent a sales share of 70% [48,49]. Additionally, the Swiss consumers can be considered frequent users of loyalty cards. Taking Coop as an example, around 80% of Coop’s annual sales were achieved with Supercard customers [50]. In this respect, Switzerland is comparatively ahead of multiple countries. In regions other than Switzerland, digital receipts are also likely to be implemented and adopted in the future due to the steadily increasing acceptance of digital payment methods, such as credit cards and mobile payment, involving an expected transition from paper to digital receipts. Thus, we believe that the results and implications from carrying out this study in Switzerland can potentially be generalized toward other regions as well.

Once users decide to donate their digital receipts to the study via the BAM service, they need to agree to multiple opt-in consent forms before their historic and future receipts can be integrated into the study (see Figure 1). To join the study, users had to opt in at least four times before their digital receipts would become part of the sample analyzed in this study. First, prospective study participants needed to be enrolled into at least one of the two Swiss loyalty card systems. Consequentially, users needed to accept the terms and conditions of at least one or even both of the loyalty card providers and opt-in toward collecting digital receipts in a digital form. Second, prospective participants needed to join the BAM service before they could participate in the study. Therefore, they needed to agree to the terms and conditions of the BAM service so that the service could retrieve their personal data from data controllers on their behalf. Third, users who already collected digital receipts from their loyalty cards needed to consent to the BAM service retrieving their digital receipts on their behalf from the loyalty card system providers directly. In one case, i.e., Migros Cumulus, this is done by linking the corresponding online account, similar to using a Facebook connect to hand over user data. In the other case, i.e., Coop Supercard, a user needs to share email-based digital receipts with the BAM service. Only then can a user’s up-to-two-year historic and new digital receipts, i.e., those that are created every time a user buys groceries and uses the loyalty card(s) at the supermarket checkout from now on, be automatically imported into the BAM service and stored in a standardized form in the personal BAM data vault. Finally, a prospective user had to join our study, which was displayed toward eligible users on the BAM service platform, and to donate their digital receipts to the study.

The study and its consent form were approved by the Ethics Committee of Swiss Federal Institute of Technology in Zurich (ETH Zurich) with the protocol code 2019-N-134 on 15 October 2019, prior to the launch of the study. In particular, the study protocol and the consent form on the BAM service required the anonymization of the donated digital receipt data. Concretely, no directly identifying personal data such as names, email addresses, phone numbers or loyalty card identifiers were shared by the BAM service with the study. To ensure the anonymity of the data donors, even the shopping locations and time of day were removed from the digital receipt dataset before the analyses conducted in this manuscript. Thus, each receipt in the final digital receipt dataset donated by the N = 464 users of the BAM service who participated in the study (see Figure 2) only contained a randomized study identifier of the respective user, the day of the year of the shopping, the shopped quantity amount, the identifier of the food item that was bought, the price, and potentially applied discounts. The possibility of re-identifying individual consumers from a maliciously acquired copy of the anonymized dataset was communicated to prospective study participants in the consent form and considered as an acceptable study risk, given the contribution of the study.

### 2.2. Food Composition Database

When using digital receipts as proxies for food choices, a common challenge is mapping food products captured via printed or digital receipts to nutrient information [20,51]. To conduct the comparisons of food shopping quality indicators, the anonymized digital receipt dataset was enriched with data about the nutritional composition of shopped food items, as digital receipts per se usually do not contain such information. The authors of this study leveraged an existing food composition database containing detailed information about over 50,000 grocery products frequently sold and consumed in Switzerland [52,53]. Driven by the recent mandates for online food nutrition databases [41], there are now trusted, curated databases (such as GS1 trustbox) as well as crowd-sourced databases (e.g., OpenFoodFacts) available to retrieve detailed nutritional information on products sold in a retail environment. This information becomes particularly useful when combined with a consumer’s shopping history.

In food composition databases, products sold in retail environments are usually identified via their global trade item number (GTIN). The GTIN is a globally unique product identifier distributed by GS1, a globally operating non-profit standards organization. Unfortunately, paper-based or digital receipts usually only contain a product’s name in terms of identifiers. Identifying a product’s GTIN from a digital receipt in fact represents one of the key challenges in digital-receipt-based monitoring. Therefore, mapping the product names to GTIN is necessary. To ensure high data quality and correct product mapping, the authors decided to conduct the product matching manually. Since both retailers, i.e., Migros and Coop, do not include the GTIN within their digital receipt formats, a heuristic was applied to correctly identify the most frequently purchased food products. In the context of this study, a total of N = 464 users were invited to donate their shopping data. These data were used to identify the most frequently occurring food products. In total, 65,391 different products that were bought using the two loyalty card systems from the two leading Swiss supermarkets were observed by assessing the entire shopping history of the N = 464 users. In total, 5950 product article identifiers from the digital receipts were mapped to corresponding GTINs. For each of the matched products, its attributes such as nutritional details (e.g., calorific energy and macro-nutrients such as protein, carbohydrate, sugar, fat, saturated fat, dietary fiber, and micro-nutrients, such as sodium, all per 100 g or mL of product; 1 g corresponds to 1 mL), its logistical data (e.g., product size in grams, kilograms, milliliters or liters), product images, allergens, and ingredients were made available for the analysis. These mapped articles correspond to 4951 of the most frequently occurring products. This is because all coupons and all non-food items were labeled identically, i.e., the study does not differentiate between different types of coupons or non-food items. These 4951 labeled product items represent most of the frequently bought food items. Since the two supermarkets also feature non-food items (e.g., plastic bags, napkins, and toilet paper) and some food items cannot be identified (e.g., ‘Menu 1’, and ‘Lunch Menu’), some frequently occurring products are not identified in this study. Neverthless, the overall matching ratio, i.e., the proportion of identified products in the shopping history of the eligible users in the four-week observation period is 69.6%. This shows that in order to capture a majority of the products purchased in digital receipts in Switzerland, less than 10% of the products must be identified in the corresponding food composition database.

The study setup described in this manuscript not only allowed for the post hoc analysis of nutritional quality of shopped food items from digital receipts within this study, but also allowed study participants to analyze the nutritional quality of their food shopping after joining the study (see Figure 3). After joining the study on the BAM service website successfully, participants gained access to a new widget that shows users the aggregated Nutri-Score [54] of their recent food shopping. For displaying and faster processing for the user experience on the BAM website, we simplified the Nutri-Score framework by using a weight-based five-letter system (A = 0.5, B = 1.5, C = 2.5, D = 3.5, and E = 4.5). For each basket, the weight-weighted average of all products was calculated and displayed on a chart, as shown in the middle of Figure 3. A careful review demonstrated that the original framework and the simplified framework yielded very similar results. The simplified version was chosen for simplicity. The analysis was provided via an application programming interface (API) and demonstrated the potential of assessing the nutritional quality of digital receipts for tailored interventions to consumers, aiming at supporting healthy food choices.

### 2.3. Food Frequency Questionnaire

After having donated their digital receipts, prospective study participants were encouraged to also complete a previously validated FFQ in order to collect objective measurement data on individual diets for the validation of the food shopping quality indicators. Multiple alternative FFQs were identified and evaluated for the study. A systematic literature research and the assessment of a meta platform that compares available FFQs (see https://www.nutritools.org/, accessed on 15 December 2021) led to the comparison of five FFQs and one web-mediated 24 h recall tool. Namely, the VioCare FFQ (See https://vioscreen.com/, accessed on 15 December 2021), the Ernährungerhebung FFQ provided by Zurich University of Applied Science (ZHAW) (see https://r2n.ernaehrungserhebung.ch/, accessed on 15 December 2021) [55], the DHQ3 FFQ (See https://www.dhq3.org/login/, accessed on 15 December 2021), the Block FFQ (see https://nutritionquest.com/login/, accessed on 15 December 2021), and the MiniMeal-Q FFQ (see https://www.nutritools.org/tools/199, accessed on 15 December 2021) were compared. Aspects such as cost, required time, Swiss aptitude, accuracy, validation and setup effort were structurally assessed. Finally, the web-mediated FFQ (see https://r2n.ernaehrungserhebung.ch/, accessed on 15 December 2021) provided by ZHAW was selected (see Figure 4), primarily due to its previous validation in Switzerland, which was also the focus region of this study [55], as well as its easy-to-use web-based administration. The FFQ-mediated questionnaire used to validate the digital receipts can be found here (link: https://gitlab.ethz.ch/jingwu/shopping-index-comparison/-/blob/master/User_Survey_and_Food_Frequency_Questionnaire__1_.pdf, accessed on 15 December 2021). The FFQ instrument was validated prior to our study and the validation study also took place in Switzerland in 2017 [55]. The validation of the FFQ allows for inferring that its dietary intake estimates correlate well with actual individual dietary intake. Thus, demonstrating the correlation between digital receipts and the FFQ allows for inferring that the digital receipts correlate with actual individual food intake. The study data, its framework and resources are in line with other validation studies in the field (i.e., HETI, GPQI, HPI). In addition, Prof. Christine Brombach, who also supervised the FFQ validation study, served as a co-author in the study, ensuring a high validity of the conducted analyses.

The chosen FFQ took an average of around 30 min for each of the N = 181 users who filled out at least 70% of the FFQ. The technical link between the BAM service and the web-based FFQ was realized via personalized links. Participants who agreed to the study consent on the BAM service received an email from BAM and were then invited via a personalized link that included a pseudonymous user identifier, linking to a user’s digital receipts as well as the user’s FFQ responses. Table 1 shows the daily food and nutrient intake of participants, based on the self-reported FFQ results.

An overview of the FFQ questions and potential answers can be found online (see https://gitlab.ethz.ch/food-coach/shopping-index-comparison, accessed on 15 December 2021). All donated data, i.e., digital receipts as well as the responses, were anonymized prior to the analyses conducted in this study. After successfully finishing these tasks, all participants received a nutritional assessment report based on their FFQ (see Figure 4) and an automatically self-updating Nutri-Score widget based on their recently shopped grocery baskets, displayed within the BAM service website (see Figure 3). In addition, participants received a financial compensation of CHF 20 (Swiss francs) (i.e., USD 21.80 (United States dollars), 28 June 2021) for completing the study.

### 2.4. Study Participants

The presented study was deployed on the BAM service online platform (see https://bitsabout.me/, accessed on 15 December 2021) and was advertised together with an invitation link through a variety of channels, including BAM’s marketplace, BAM’s newsletter, social media and billboards on the local university campus as well as university shuttle buses. This approach of using BAM’s marketplace and the university network ensured to address younger as well as more mature households, as the socio-demographic characteristics on the BAM website represent more mature consumers (see Table 2).

Participants were recruited from 22 December 2018 to 10 June 2021. Participants used their own devices (e.g., laptops and/or mobile phones) to enroll in the study. The relatively high barriers to join the study, as shown in Figure 1, led to a slow uptake of participants. Hence, the participant recruitment was conducted on a rolling basis of over two years in order to collect the required sample size for a robust comparison of the suggested food shopping quality indicators.

To ensure completeness and comparability between the FFQ data and the digital receipt data, participants were required to meet several eligibility criteria, which are illustrated in Figure 2. In total, N = 464 participants joined the study via accepting the study consent form that was displayed on the BAM service website. Out of those, N = 231 followed the email-based invitation to also start the web-based FFQ (i.e., email response rate of 50.2%). Fifty participants were excluded because they did not complete the FFQ (i.e., drop out rate 21.6%). To ensure that food shopping recorded on loyalty cards is representative of actual food intake, eligible users should shop at least 1/7 of the estimated calorific requirement of all people who share the loyalty cards with participants, within the four weeks preceding their FFQs. The estimated energy intake of a person who is ≥13 years old was 2250 kcal/day [56], irrespective of gender. That of a child (<13 years old) was estimated to be 0.575 [57] times the energy requirement of a person who is older than 13, i.e., 2250 kcal/day, irrespective of gender. The four-week time window was defined by the FFQ, which was validated to estimate a participant’s typical diet on a four-week basis [55]. Consequently, 91 participants were excluded, as they did not meet the criteria (exclusion rate 50.2%, i.e., 91 from 181 users who completed their FFQ and donated digital receipts). Finally, one participant self-reported his/her own BMI to be 109.8 kg/m^2^ (height, 88 cm; weight, 85 kg) and was therefore excluded. The final dataset included 89 users with a completed FFQ and an acceptable amount of shopped products captured by digital receipts in the four weeks prior to the FFQ.

On average, an eligible participant was 36.2 years of age (standard deviation (SD): 10.3) and had an average BMI of 24.4 (SD: 3.8). Compared to the general Swiss population, the recruited sample (see Table 2) has a higher male to female ratio (76.4% in the sample vs. 49.2% in Switzerland), a lower average BMI (24.4 in the sample vs. 25.3 in Switzerland), and a lower average age (36.2 years in the sample vs. 43.1 in Switzerland). With regard to the observed shopping behavior, participants in the final dataset spent on average a total of CHF 230.30 (SD: CHF 175.60) on an average of 39.9 kg (standard deviation: 32.1 kg) grocery products over the four weeks before finishing the FFQ. The average number of adults and children who were sharing their loyalty cards are 1.7 (SD: 1.0) and 0.5 (SD: 0.9) respectively (see Table 3). In terms of individual diet as determined by the FFQ, the N = 89 users on average consumed 1.17 (SD: 1.2) portions of meat per day (see Table 1). Similarly, they consumed on average 2.55 (SD: 1.71) portions of vegetables and 1.38 (SD: 1.16) portions of fruits per day. Hence, the average participant did not reach the publicly recommended three portions of vegetables and two portions of fruits per day [58]. The participants consumed on average 0.32 (SD: 0.33) portions of whole grains per day. Finally, the participants consumed an average of 2.93 (SD: 1.95) portions of sweets per day. The fact that the Swiss population consumes too many sweets has been observed in the annual food intake study MenuCH for many years [59,60]. In terms of nutritional intake, the N = 89 participants on average consumed 2.1 g (SD: 1.5) of sodium, 27.1 (SD: 14.3) grams of dietary fibers, 37.5 (SD: 26.0) grams of saturated fatty acids and an estimated amount of 10.4 (SD: 8.5) grams of added sugar per day. Compared to the annual food intake study MenuCH, these values seem representative of typical dietary intake and in line with the observations for the dietary behavior of the Swiss population [59,60].

### 2.5. Validation and Comparison of Food Shopping Quality Indicators

To assess the calibration and discrimination capacity of the alternative food shopping quality indicators, five different indicators, namely FSA-NPS DI, GPQI, HEI-2015, HETI and HPI, were selected and calculated. Their respective calibration and discrimination performance for individual N = 89 dietary behavior was computed. We selected the indicators for multiple reasons. First, all five indicators were defined to represent a quantitative indication of an individual’s general dietary habitual patterns. Second, the indicators can all be calculated using the digital receipts mandated by GDPR [40] as well as the food information, as mandated for products sold online by EU1169 [41]. Therefore, these indicators have the potential to support millions of consumers in the European Union in terms of monitoring nutritional quality using digital receipts. Finally, the five indicators have not yet been cross-validated in the same geographic region. While the GPQI and HEI-2015 were defined in the US, HETI was defined in Australia, and the HPI was defined in France. Further, the FSA-NPS DI was defined in the United Kingdom (UK) and later adopted in France. As argued earlier, Switzerland represents a suitable study context for such a validation study.

All five indicators were be calculated on a four-week basis, as the FFQ was previously validated over the course of a four-week time period [55]. Receipts whose timestamps were within the four weeks prior to the completion of the FFQ were used for calculation. The original definitions of the FSA-NPS DI, GPQI, HETI and HPI were followed and not adapted in terms of calculation. As discussed in the previous chapter, the HEI-2015 [44] was adopted to digital receipts, similar to how the Healthy Eating Index-2010 (HEI-2010) was adapted to food shopping [43]. As defined in their publications, the GPQI, HEI-2015, HETI and HPI are primarily based on health-relevant food groups. The HEI-2015 also includes some nutrients but not the other three food shopping quality indicators. Thus, for their calculation, the authors included only items belonging to the relevant food groups used in their respective definitions. When calculating the FSA-NPS DI, all shopped food items were included.

Results from the FFQ were taken as the objective measurements. We followed the definition of the Diet Quality Index [61], a Swiss dietary index, and aggregated food to five groups, namely meat and meat products; vegetables and salad; fruits; wholegrain products; and sweets, salty snacks, sugar-sweetened beverages and alcohol. On the nutrient level, we retrieved the consumption of sodium, dietary fiber and saturated fatty acids directly, using the food composition database. As added sugar content is not mandated in Europe [41], we estimated it based on an established approach proposed by Louie et al. [62].

To validate the indicators with the nutritional intake as determined by the FFQ, certain conditions were followed to ensure a coherent process. First, digital receipts that could not be identified (e.g., rare products) and therefore have missing nutritional data, were discarded. To assess the number of portions eaten by each individual user as determined by the FFQ, or shopped as determined by the digital receipts, the following considerations were agreed upon. First, to calculate the number of consumed meat portions, dried meat portions were defined as 30 g per portion, while non-dried meat as 120 g per portion. Second, to assess the number of fruit portions shopped or eaten, 30 g of dried fruits were defined as one portion, while 120 g of fresh or frozen fruits or 200 mL of fruit juices were defined as one portion. Third, 30 g of dried vegetables accounted for one portion, while 120 g of fresh or frozen vegetables or 200 mL of vegetable juices were defined as one portion. The number of portions of whole grains eaten or shopped were defined as follows: 100 g for bread, 30 g for cereals or flakes, 60 g for crisp bread and 30 g for cereals. In this study, whole grains were defined as grain products that include at least 5 g of dietary fiber per 100 g of product [63,64,65]. Finally, for sweets, the definitions of one portion were set to: 17 g for chocolate, 20 g for cocoa products and jams, 30 g for bonbons, cereal bars and cookies, 50 g for pudding and ice creams, 120 g for sweet cakes and similar. A detailed overview of the portion sizes used in the data processing for the FFQ and digital receipts can be found in the datasets that were published together with this manuscript.

Based on the obtained data from the participants’ FFQs and digital receipts, we assessed the *calibration capacity* and the *discrimination capacity* of the considered food shopping quality indicators. To evaluate the *calibration capacity* of the models, Pearson correlation coefficients between food shopping quality indicators and *absolute*/*relative* nutritional intake were calculated. In this context, higher correlation coefficients represent higher calibration capacity. More specifically, a high correlation of a food shopping quality indicator with a health-relevant nutrient or food group underlines a valid calibration of the food shopping quality indicator for the nutrient or food group respectively.

To evaluate the *discrimination capacity* of the proposed food shopping quality indicators, the nutritional intake of three compliance tertiles were compared. Similar to validation studies of single food shopping quality indicators that divided their sample into three segments, tertile T1 maps to the lowest compliance to a given indicator, and T3 maps to the highest compliance. For the assessment of the discrimination capacity of the indicators, their potential to discriminate all three tertiles T1–T3, as well as their ability to distinguish pairwise differences between each combination of the tertiles were evaluated. In total, four statistical tests, namely one Kruskal–Wallis test for the three tertile comparisons (T1, T2, and T3) and three Mann–Whitney U tests for the pairwise comparisons (T1 vs. T2; T2 vs. T3; and T1 vs. T3, respectively) were conducted for each combination of an indicator with a nutritional intake category as determined by the FFQ. A significantly different result is counted as one point, while a non-significant result is counted as zero point. The total points of each indicator were obtained by summing up the points across all comparisons where the respective indicator was involved. Therefore, more points represent better discrimination capacity, and the maximum achievable number of points is 36.

## 3. Results

We obtained the following results on the *calibration* and *discrimination* capacities of the investigated indicators.

### 3.1. Calibration Capacity: Correlations between Food Shopping Quality Indicators and Nutritional Facts

As shown in Table 4, food shopping quality indicators and individual *absolute* daily nutritional intake were in general weakly to moderately correlated. Among all food shopping quality indicators, only FSA-NPS DI was weakly correlated with the highest number (4) of food and nutrient categories, i.e., the Pearson’s correlation coefficient was between 0.1 and 0.3. The strongest correlation was observed between HEI-2015 and *absolute* dietary fibers. Generally, the correlations were significantly stronger when using individual daily density-based *relative* nutritional intake, as shown in Table 5. The strongest correlation was observed between FSA-NPS DI and *relative* intake of dietary fibers, with a Pearson’s r of 0.500. Compared to the other four indicators, the FSA-NPS DI was correlated more strongly with the nutritional facts on both *absolute* and *relative* scales, demonstrating the highest correlation coefficient for four (six) out of the nine dimensions for *absolute* (*relative*) dietary intake captured from FFQs. On the other hand, the correlations between HETI, HPI and nutritional facts were in general weaker than those among other food shopping quality indicators and nutritional facts, no matter on the *absolute* or *relative* scale. On average, the five food shopping indicators calibrate fruits and dietary fibers the best, on both *absolute* or *relative* scales.

### 3.2. Discrimination Capacity: Comparisons of Nutritional Facts across Compliance Tertiles

Table 6 and Table 7 show the assessment results of the indicators’ *discrimination capacity*, differentiating *absolute* and *relative* nutritional intake, respectively. Regarding the *absolute* food category intake as shown in Table 6, all indicators were able to differentiate participants’ intake of fruits to a certain degree. On the contrary, no indicator managed to differentiate participants’ added sugar intake, which is not declared on products in Switzerland, but was only indirectly estimated using products’ category affiliation and sugar content [62], in contrast to the other nutrients and food groups. In general, the FSA-NPS DI outperformed the other indicators in differentiating participants’ *absolute* nutritional intake, as 17 out of the 36 statistical tests were significant (p<0.05). It was the only indicator that was capable of differentiating participants’ intake of sweets, salty snacks, sugar-sweetened beverages, alcohol, sodium and saturated fatty acids well. All these aspects are important health-influencing factors. When it comes to differentiating the *relative* food category intake, as shown in Table 7, all indicators, except the HPI, performed better compared to distinguishing *absolute* food intake, yielding a higher number of significant results. Moreover, the FSA-NPS DI outperformed the other indicators, demonstrating 19 significant results out of 36 statistical tests. HEI-2015 and HETI performed equally well when the nutritional intake was assessed on a *relative* basis, having the same number of significant results. However, the corresponding details were different. For instance, HEI-2015 was able to differentiate the *relative* sodium intake, while HETI was not. Note that when considering specific food categories or nutrients, the best performance might not always be achieved by the same indicator. On average, the five food shopping indicators discriminate fruits and dietary fibers robustly on both *absolute* or *relative* scales.

To confirm the ability of the FSA-NPS DI to distinguish users with different dietary habits, the characteristics of the highest, medium and lowest tertiles of the inverted FSA-NPS DI were assessed in Table 8 and Table 9, which shows the *absolute* and *relative*, i.e., per 1000 kcal, food intake of the entire study sample and three tertiles as measured by the FSA-NPS DI. The inverted FSA-NPS DI was used to make it more comparable to other food shopping quality indicators. The higher the inverted FSA-NPS DI, the healthier the food shopping. T3 has the shopping baskets of the healthiest nutritional quality. The median and interquartile ranges (IQRs) were reported because of the non-normal distribution of the inverted FSA-NPS DI. On both *absolute* and *relative* scales, T1 consumed significantly more meat, less fruit and less dietary fiber, compared to T2 and T3.

## 4. Discussion

### 4.1. Summary

This manuscript represents the first ever quantitative validation study on the *calibration* and *discrimination* ability of previously suggested food shopping quality indicators, including FSA-NPS DI, GPQI, HEI-2015, HETI and HPI. In total, digital receipts from two loyalty card systems and validated FFQs from N = 89 individuals in Switzerland were collected to validate five indicators that were previously developed and validated in separate regions in the world.

As shown in Table 4 and Table 5, the assessed indicators correlate weakly to moderately with *absolute* and slightly more strongly with *relative* individual daily nutritional intake. The following reasons might explain why the correlations are only weak to moderate. First, there is a time lag between food shopping and eating. For instance, a person bought fruits on 31 January but consumed them in the first week of February. Second, inter-day and inter-week variances can be high and are commonly present. For example, a person can eat pizza on a certain day and skip lunch on the next day. A person can also fast for some time for different reasons. Physical activity also influences how much energy a person needs [66]. Third, only 69.6% of receipts were successfully matched. With higher product detection rates, the correlation of digital-receipt-based purchase indicators and individual dietary intakes might even be higher. Fourth, the relationship of food purchases and individual dietary intake are moderated by hard-to-assess behavioral factors, such as food waste, food processing, and out-of-home eating. Still, the observed correlations are in line with the literature and acceptable in comparison with bio-sampling (e.g., sodium excretion or blood sampling) and even 24 h recalls [67]. Although the purchase indicators have a lower accuracy than contemporary diet intake assessment methods, they are more cost-effective and scalable. Thus, they are still qualified as either stand-alone or complementary indicators for large-scale diet monitoring and long-term dietary behavior change interventions.

As shown in Table 4, Table 5, Table 6 and Table 7, the performance of indicators was generally better in calibrating and discriminating the calorie-weighted *relative* individual dietary intake than the *absolute* individual dietary intake. This could be because when both using *relative* scales, the gap between individual calorie-weighted food intake and household expenditure- or calorie-weighted food shopping is narrower. The FSA-NPS DI has the best *calibration capacity* and *discrimination capacity*, followed by HEI-2015. The relatively inferior performance of GPQI, HETI and HPI might be due to the following reasons. First, calorie-weighted approaches likely perform better than expenditure-weighted approaches in the nutrition context. This is intuitive since inflation, prices and promotions moderate the amounts consumed from food products that are similar in their nutritional composition but different in price. Even differences in currency exchange rates moderate the performance of expenditure-weighted indicators, as the GPQI, HETI and HPI are designed and calibrated in the US, Australia and France respectively by evaluating shares of wallet (in USD or Australian dollars or in Euros). A transformation to other currency regions is likely to inhibit the performance of such expenditure-based indicators. Hence, weight-based or calorie-based weighted indicators are likely to correlate better with individual food intake, as they measure the representation of food choices better from a dietary perspective. Therefore, if available, data on consumed products’ respective weights and their calorific impact should be taken into account when designing purchase quality indicators.

The study does not give reason to believe that the geographic context in which an indicator was designed limits its performance abroad. While the FSA-NPS DI was developed in the UK, the GPQI and HEI-2015 were defined in the US, HETI was defined in Australia, and the HPI was defined in France. Since this validation study was conducted in Switzerland, it might be surprising that although Switzerland is closer to France, where the HPI was developed, than to the US, where the HEI-2015 was developed, HEI-2015 performed better than HPI in calibrating and discriminating the dietary intake of a Swiss study sample. In contrast, the FSA-NPS DI, which was developed in the UK, performed best. Hence, no structural dominance of European indicators could be observed. These results therefore indicate that food shopping quality indicators are indeed transferable to a new region, such as Switzerland and can, in fact, give a relatively reliable indication of the nutritional quality of individual diets by assessing their digital receipts. The FSA-NPS DI consistently outperformed the other food shopping quality indicators in calibrating and differentiating participants’ nutritional intake, on both *absolute* and *relative* density-based scales. Hence, a validation of the FSA-NPS DI within digital receipts outside Europe could be an interesting foray for future research.

The results suggest that considering both food groups and nutrients, particularly fruits and dietary fibers, might be important in the design of food shopping quality indicators. The GPQI, HETI and HPI do not include any nutrient categories in their definitions. On the contrary, the HEI-2015 explicitly includes fatty acids, sodium, added sugars and saturated fats, and the FSA-NPS DI includes sodium, saturated fats, sugar and dietary fiber. These two indicators have better performance in calibrating and discriminating actual dietary intake. As shown in Table 4, Table 5, Table 6 and Table 7, dietary fiber can generally be calibrated and discriminated the best by the food shopping indicators. Therefore, dietary fiber seems to be an important factor to consider in designing and choosing relevant purchase indicators. In terms of relevant food groups, all indicators can calibrate and discriminate fruits well. These results support the important roles of fruits in a healthy diet and in the design of food shopping indicators. Th European Commission also recommends its member states to track their consumers’ intake of fruits within their European Core Health Indicators (ECHIM) list of health-relevant policy indicators [68].

### 4.2. Contribution

This study has multiple contributions to research and practice. First, this study represents the first validation of previously suggested food shopping quality indicators from different regions in the world within a thorough comparative quantitative assessment. Second, the consistent out-performance of the FSA-NPS DI in nutritional intake calibration and discrimination demonstrates that it could be a good choice for general purpose of use, particularly when developing tools for long-term diet monitoring and intervention. Third, the study found that fruits and dietary fibers are better calibrated and discriminated by food shopping indicators. This suggests the important roles of fruit and dietary fibers in a healthy diet. It might be useful to consider these two aspects when designing a food shopping indicators. Added sugar intake was not well captured by the assessed food shopping quality indicators. This finding suggests that there is still room for the development of different purpose-specific food shopping quality indicators (e.g., estimating the risk for diabetes type two by estimating intake levels of added sugar and carbohydrate quality). Fourth, researchers, practitioners and designers for food choice monitoring systems and behavior change interventions should take away that these indicators capture the calorie-adjusted *relative* nutritional intake better than *absolute* dietary intake. Finally, the study demonstrates that regulatory mandates such as the GDPR [40] and the mandate for nutritional declaration for food products sold online (EU)1169 [41] can pave the way for novel tools for scalable, non-invasive monitoring of food choices and tailored digital behavior change interventions, using digital receipts and food composition databases.

### 4.3. Limitations

This study possesses limitations. We select for discussion the ones which warrant special attention. First, the number of participants restricts the generalizability of the results. Compared with the actual composition of the Swiss population [69,70], the female participation ratio, the average age, and the mean BMI of the present sample were lower. Multiple factors could have led to the sample not being representative. First, most users on the BAM platform are male. Second, participants were mainly health-conscious individuals in or beyond tertiary education, considering where our advertisements were placed (e.g., university shuttle buses and billboards). In addition, the study is biased toward loyalty card holders of the leading two retailers in Switzerland since they are the only group that is able to participate in the study. Compared to non-card-holders, they might have slightly different eating or food shopping habits.

Regarding the FFQ, although the FFQ was validated previously in Switzerland, the tool has certain limitations, such as potentially missing food items in the questionnaire or the fact that vitamin and mineral supplements, which can be very important for vegetarians, are not covered in the FFQ. All these might lead to reporting biases, which are commonly present [71].

The five food shopping quality indicators were calculated using *relative* expenditure share or portion share per 1000 kcal. This implicitly assumed that each individual consumes the same proportion of food across all food categories, regardless of gender and age. This assumption could be biased. For instance, if a participant is a vegan but he or she lives with meat eaters, we assume that this participant consumes meat as well if there is meat in the receipts. Moreover, the food waste proportion in different food categories tends to be different. As the goal is to check if digital receipts are able to provide indications about dietary behavior, but not to predict actual food intake, these problems are rather mild for now.

Furthermore, inaccurate or missing information in the food composition database forms another limitation of the study. The manual matching process between article identifiers in digital receipts and their corresponding GTINs is challenging and can sometimes lead to ambiguous results. To mitigate this limitation, we invested a significant effort in the food composition database and manually mapped the most frequent 5950 products’ article identifiers, which is far beyond the top 3000 products mapped and identified in similar studies.

Lastly, the amount of added sugar captured by the digital receipts seems low compared to the sugar content, which might be due to different food category definitions. As the declaration of added sugar is not mandated, we used an estimation based on sugar content and category affiliation of shopped products [62]. While the food categorization frameworks between studies might differ and were defined in regions other than Switzerland, this effect might have caused an underestimation of added sugar in this study. Coherent category definitions seem pivotal, as we found that the performance results of the GPQI and HPI indicators are very sensitive to changes in the definition of refined grains, for example.

### 4.4. Future Work

First and foremost, it is essential for the future of this and related studies to recruit a larger and more representative sample. In addition, data quality of digital receipts and food composition databases is another important factor that constructs gaps in research and practice. The authors have been and will continue being dedicated to increasing the matching ratio of digital receipts. To overcome the limitations of the FFQ, bio-sampling (e.g., sodium excretion or blood sampling) could be added to offset the potential of reporting biases, but it means higher costs as well. To capture information that is not in receipts, conducting a survey about other aspects of food behavior, such as food waste, might be necessary. To ensure the reproducibility of similar studies in the future, researchers should adhere to established food classification schema, such as FoodEx2 (revision 2) [72]. Although FoodEx2 is only defined for the European context, such classification schema can be very helpful in transferring concepts to other regions as well. Ultimately, a global food classification schema would be helpful to reproduce the suggested food shopping quality indicators internationally.

To support the extensive efforts required for the correct mapping of digital receipts and food composition databases, machine-learning based algorithms [73] can support the (semi-)automatic correction of errors in food composition databases [74] or the correct identification of products present in digital receipts (e.g., word2vec) [75,76].

This study also calls for the integration of product identifiers into digital receipt standards. Currently, retailers are not required to integrate relevant product identifiers, such as the GTIN, into their digital receipt structures. Therefore, the identification of text-based product identifiers within food product composition databases requires lots of manual effort. It would, therefore, be beneficial for the development of scalable digital receipt-based food choice monitoring and interventions if regulators would extend the right for data portability as mandated by the GDPR [40] and mandate the use of product identifiers within digital receipt standards.

In addition to the FSA-NPS DI, future research could assess the potential of superior food shopping indicators for estimating or even predicting the nutritional intake and health states of consumers. Finally, future regulatory mandates, such as the expected GDPR 2.0 in the European Union, could enforce the use of standardized, real-time APIs to further ease the barriers for data portability of personal datasets including shopping records.

## 5. Conclusions

In this study, we present the first comparison of the calibration and the discrimination capacities of shopping indicators FSA-NPS DI, GPQI, HEI-2015, HETI and HPI on a dataset that was obtained from 89 participants and included responses to an FFQ and real grocery shopping data using digital receipts collected from two loyalty card systems in Switzerland. Our results show that, overall, the surveyed indicators correlate only weakly to moderately with *absolute* and slightly more strongly with *relative* individual daily nutritional intake. All indicators are weakly to moderately suitable to differentiate health-relevant daily nutritional intake behavior using digital receipts. Among these indicators, the FSA-NPS DI in general outperforms the other indicators in our dataset, having the best calibration and discrimination ability, thus contributing an empirically validated guideline regarding the selection of shopping indicators. This is counterintuitive, as the FSA-NPS DI was primarily designed to discriminate the basket quality, but not the compliance to certain dietary guidelines nor to certain food group intake. The relatively inferior calibration performance of the GPQI, HETI and HPI might be because they were designed and validated by evaluating shares of wallet (in Australian dollars or in euros) in the US, Australia and France, rather than assessing nutrients or food groups by their weight or calorie contribution. This might make it inappropriate to translate them to other regions, as the prices and retail market dynamics in the US, Australia and France might differ from those in other countries. Adopting these indicators might require a re-calibration within other regions. In the future, we plan to enlarge the sample size, improve the data quality, and rerun the analysis using machine learning frameworks to estimate individual dietary intake deficits from digital receipts more accurately. In addition, we plan a further investigation into the reasons behind the differences in the calibration and discrimination capacities of the surveyed indicators in the future.

## Figures and Tables

**Figure 1 nutrients-14-00159-f001:**
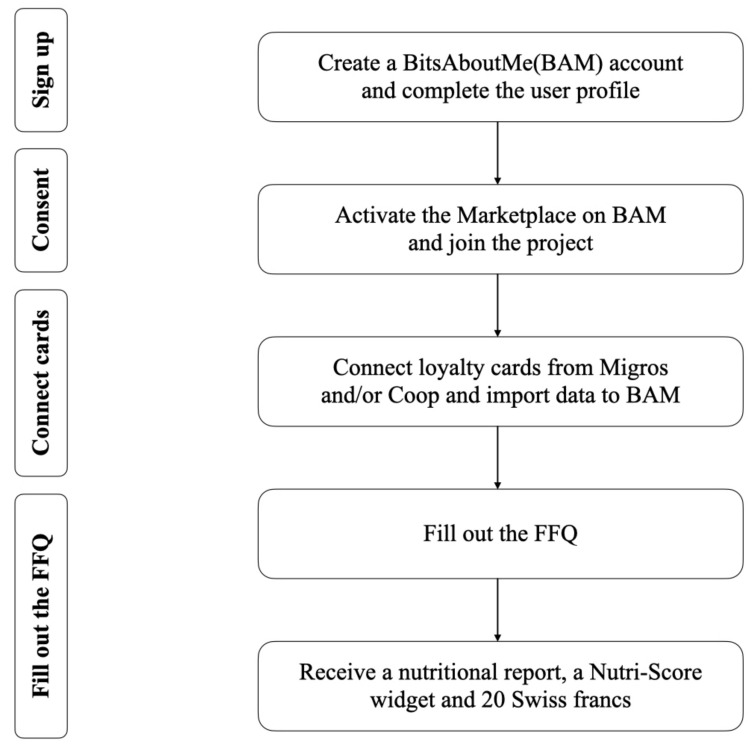
Participant on-boarding flow.

**Figure 2 nutrients-14-00159-f002:**
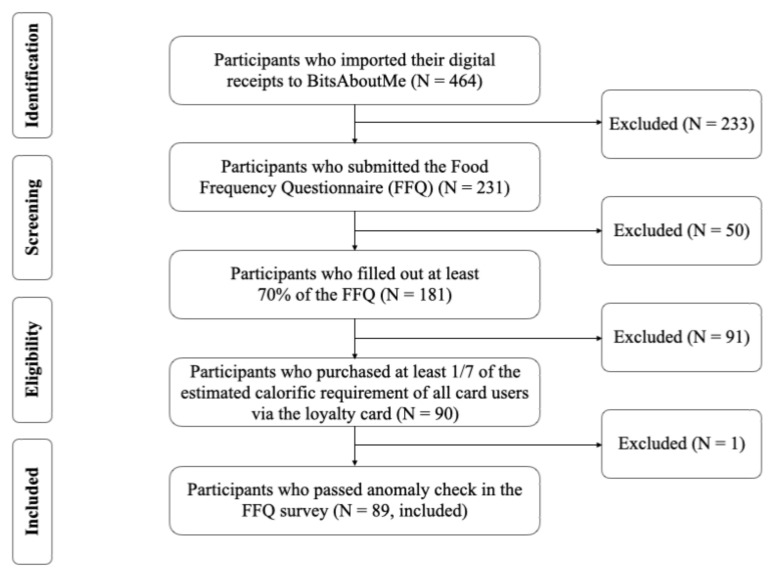
The procedure of excluding ineligible participants.

**Figure 3 nutrients-14-00159-f003:**
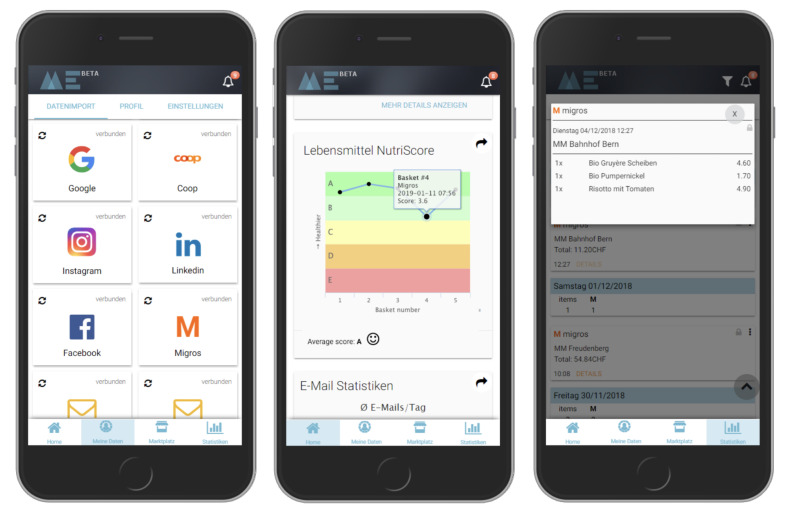
Enriching digital receipts with food composition data and displaying the weight-averaged Nutri-Scores of aggregated baskets.

**Figure 4 nutrients-14-00159-f004:**
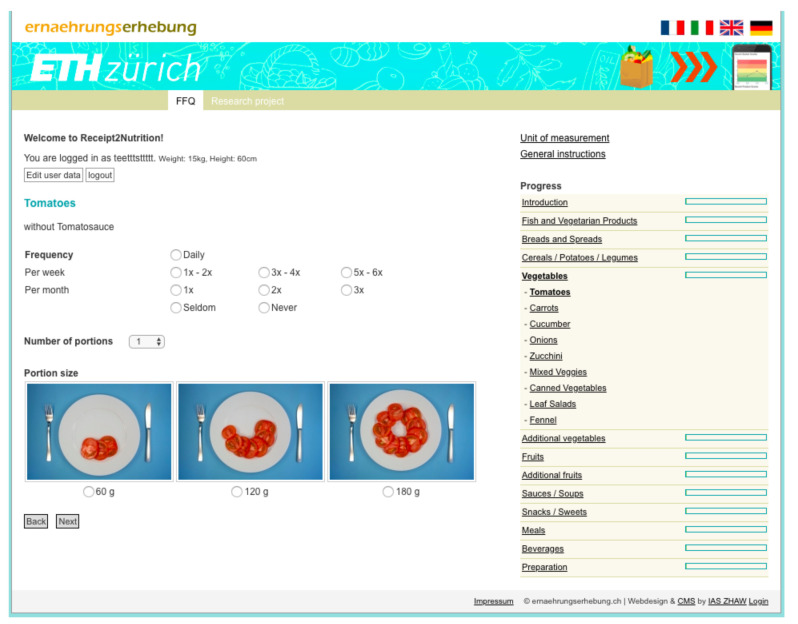
Web-mediated food frequency questionnaire (FFQ).

**Table 1 nutrients-14-00159-t001:** *Absolute* individual daily nutritional intake, N = 89.

Category	Mean	Standard Deviation
*Unit: portions/day*		
Meat and meat products	1.17	1.20
Vegetables and salad	2.55	1.71
Fruits	1.38	1.16
Whole grain products	0.32	0.33
Sweets, salty snacks, sugar-sweetened beverages, alcohol	2.93	1.95
*Unit: grams/day*		
Sodium	2.1	1.5
Dietary fibers	27.1	14.3
Saturated fatty acids	37.5	26.0
Added sugar	10.4	8.52

**Table 2 nutrients-14-00159-t002:** Demographic summary of participants.

Sample	Count (%)
*Gender*	
Male	68 (76.4)
Female	21 (23.6)
Other	0 (0.0)
*Age [yrs]*	
18–29	29 (32.6)
30–39	29 (32.6)
40–49	18 (20.2)
> 50	13 (14.6)
*Body Mass Index [kg/m^2^]*	
Underweight (<18.5)	2 (2.3)
Normal (≥18.5 and <25.0)	55 (61.8)
Overweight (≥25.0 and <30.0)	22 (24.7)
Obese (≥30)	10 (11.2)
Total	89 (100.0)

**Table 3 nutrients-14-00159-t003:** Food shopping characteristics of the study sample, N = 89.

Characteristics of Observed Food Shopping Behavior ^a^	Mean (SD ^b^)
*Household*	
Adults sharing the loyalty card(s)	1.7 (1.0)
Children sharing the loyalty card(s)	0.5 (0.9)
*Food shopping quantity identified via digital receipts*	
Amount spent in Swiss francs (CHF)	230.30 (175.60)
Amount spent in United States dollars (USD) ^c^	250.28 (190.83)
Weight ^d^ of shopped food products in kg	39.9 (32.1)

${}^{a}$ This is based on the food shopping data in the four weeks before finishing FFQs. ${}^{b}$ SD: standard deviation. ${}^{c}$ Conversion CHF/USD on date: 30 June 2021. ${}^{d}$ Conversion for liquids: 1 g = 1 mL.

**Table 4 nutrients-14-00159-t004:** Pearson correlation coefficients between food shopping quality indicators and *absolute* individual daily nutritional intake, N = 89.

Indicators ^a^	- FSA-NPS DI ^b^	GPQI ^c^	HEI-2015 ^d^	HETI ^e^	HPI ^f^
*Unit: portions/day*					
Meat and meat products	**−0.246** *	0.000	−0.083	−0.099	−0.060
Vegetables and salad	**0.235** *	0.140	0.190	0.181	0.177
Fruits	0.239 *	0.215 *	0.254 *	0.274 **	**0.288 ****
Wholegrain products	0.161	0.184	**0.322** **	0.232 *	0.135
Sweets, salty snacks, sugar-sweetened beverages, alcohol	−0.111	−0.036	**−0.124**	−0.026	−0.002
*Unit: grams/day*					
Sodium	**−0.121**	0.050	0.023	0.072	0.027
Dietary fibers	0.312 **	0.178	**0.329** **	0.296 **	0.173
Saturated fatty acids	**−0.144**	0.033	0.006	0.034	−0.008
Added sugar	−0.101	**−0.197**	−0.193	−0.137	0.138
Points	**4**	1	3	0	1

${}^{a}$ The highest absolute value is marked bold to indicate the best calibrated food shopping quality indicator for each nutrient or food group. ${}^{b}$ - FSA-NPS DI: Inverted Food Standards Agency Nutrient Profiling System Dietary Index. We inverted the original FSA-NPS DI scores to make them directly comparable to other food shopping quality indicators. The higher the inverted FSA-NPS DI, the healthier the food shopping. ${}^{c}$ GPQI: Grocery Purchase Quality Index-2016. ${}^{d}$ HEI-2015: Healthy Eating Index-2015. ${}^{e}$ HETI: Healthy Trolley Index. ${}^{f}$ HPI: Healthy Purchase Index. * *p* < 0.05. ** *p* < 0.01.

**Table 5 nutrients-14-00159-t005:** Pearson correlation coefficients between food shopping quality indicators and *relative* individual daily nutritional intake, N = 89.

Indicators ${}^{a}$	- FSA-NPS DI ${}^{b}$	GPQI ${}^{c}$	HEI-2015 ${}^{d}$	HETI ${}^{e}$	HPI ${}^{f}$
*Unit: portions/1000 kcal*					
Meat and meat products	**−0.359** ***	−0.061	−0.241	−0.240 *	−0.090
Vegetables and salad	**0.321** **	0.136	0.191	0.108	0.133
Fruits	**0.354** ***	0.195 *	0.238 *	0.197	0.245 *
Wholegrain products	0.231	0.101	**0267** *	0.193	0.080
Sweets, salty snacks, sugar-sweetened beverages, alcohol	−0.097	−0.068	**−0.197**	−0.139	0.069
*Unit: g/1000 kcal*					
Sodium	**−0.244** *	−0.092	−0.178	−0.055	−0.045
Dietary fibers	**0.500** ***	0.126	0.342 **	0.235 *	0.139
Saturated fatty acids	**−0.367** ***	−0.125	−0.228 *	−0.177	−0.143
Added sugar	−0.093	**−0.329 ****	−0.316 **	−0.306 **	−0.259 *
Points	**6**	1	2	0	0

${}^{a}$ The highest absolute value is marked bold to indicate the best calibrated food shopping quality indicator for each nutrient or food group. ${}^{b}$ - FSA-NPS DI: Inverted Food Standards Agency Nutrient Profiling System Dietary Index. We inverted the original FSA-NPS DI scores to make them directly comparable to other food shopping quality indicators. The higher the inverted FSA-NPS DI, the healthier the food shopping. ${}^{c}$ GPQI: Grocery Purchase Quality Index-2016. ${}^{d}$ HEI-2015: Healthy Eating Index-2015. ${}^{e}$ HETI: Healthy Trolley Index. ${}^{f}$ HPI: Healthy Purchase Index. * *p* < 0.05. ** *p* < 0.01. *** *p* < 0.001.

**Table 6 nutrients-14-00159-t006:** Discrimination potential of relevant food shopping quality indicators to differentiate health-relevant individual daily nutritional intake ${}^{a}$ behavior (*absolute*, i.e., weight-based), N = 89.

Indicator	- FSA-NPS DI ${}^{b}$	GPQI ${}^{c}$	HEI-2015 ${}^{d}$	HETI ${}^{e}$	HPI ${}^{f}$
*Unit: portions/day*					
Meat and meat products	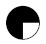	○	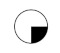	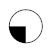	○
Vegetables and salad	○	○	◓	○	○
Fruits	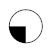	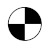	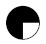	◕	◕
Wholegrain products	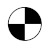	○	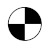	◕	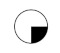
Sweets, salty snacks,sugar-sweetened beverages,alcohol	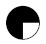	○	○	○	○
*Unit: grams/day*					
Sodium	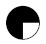	○	○	○	○
Dietary fiber	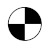	○	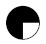	◕	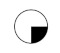
Saturated fatty acids	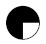	○	○	○	○
Added sugar	○	○	○	○	○
Points	**17**	2	11	10	5

a The products contained in each food category can be found on https://gitlab.ethz.ch/food-coach/shopping-index-comparison, accessed on 15 December 2021. b - FSA-NPS DI: Inverted Food Standards Agency Nutrient Profiling System Dietary Index. We inverted the original FSA-NPS DI scores to make them directly comparable to other food shopping quality indicators. The higher the inverted FSA-NPS DI, the healthier the food shopping. c GPQI: Grocery Purchase Quality Index-2016. d HEI-2015: Healthy Eating Index-2015. e HETI: Healthy Trolley Index. f HPI: Healthy Purchase Index. ○ No test was significant. 
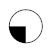
 The Mann–Whitney U test between the 1st and the 3rd tertiles was significant (*p* < 0.05). 
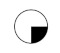
 The Mann–Whitney U test between the 1st and the 2nd tertiles was significant (*p* < 0.05). ◓
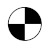
◕
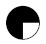
 Multiple above-mentioned statistical tests were significant (*p* < 0.05).

**Table 7 nutrients-14-00159-t007:** Discrimination potential of relevant food shopping quality indicators to differentiate health-relevant individual daily nutritional intake ${}^{a}$ behavior (*relative*, i.e., calorie-adjusted), N = 89.

Indicator	- FSA-NPS DI ${}^{b}$	GPQI ${}^{c}$	HEI-2015 ${}^{d}$	HETI ${}^{e}$	HPI ${}^{f}$
*Unit: portions/1000 kcal*					
Meat and meat products	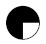	○	◕	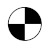	○
Vegetables and salad	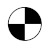	○	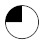	○	○
Fruits	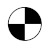	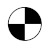	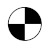	○	◑
Wholegrain products	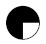	◕	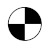	◕	○
Sweets, salty snacks,sugar-sweetened beverages,alcohol	○	○	○	○	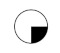
*Unit: g/1000 kcal*					
Sodium	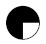	○	◑	○	○
Dietary fibers	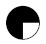	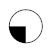	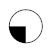	◕	○
Saturated fatty acids	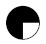	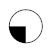	○	◕	○
Added sugar	○	○	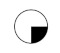	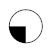	○
Points	**19**	7	12	12	3

a The products contained in each food category can be found on https://gitlab.ethz.ch/food-coach/shopping-index-comparison, accessed on 15 December 2021. b - FSA-NPS DI: Inverted Food Standards Agency Nutrient Profiling System Dietary Index. We inverted the original FSA-NPS DI scores to make them directly comparable to other food shopping quality indicators. The higher the inverted FSA-NPS DI, the healthier the food shopping. c GPQI: Grocery Purchase Quality Index-2016. d HEI-2015: Healthy Eating Index-2015. e HETI: Healthy Trolley Index. f HPI: Healthy Purchase Index. ○ No test was significant. 
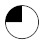
 The Mann–Whitney U test between the 2nd and the 3rd tertiles was significant (*p* < 0.05). 
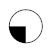
 The Mann–Whitney U test between the 1st and the 3rd tertiles was significant (*p* < 0.05). 
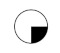
 The Mann–Whitney U test between the 1st and the 2nd tertiles was significant (*p* < 0.05). ◑
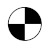
◕
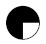
 Multiple above-mentioned statistical tests were significant (*p* < 0.05).

**Table 8 nutrients-14-00159-t008:** Median and interquartile range (IQR) of the *absolute* nutritional intake across the tertiles of the inverted Food Standards Agency Nutrient Profiling System Dietary Index (-FSA-NPS DI) ${}^{a}$.

	-FSA-NPS DI Score Tertile							
	Overall (N = 89)	T1 (N = 30)	T2 (N = 29)	T3(N = 30)				
	Median (IQR)	Median (IQR)	Median (IQR)	Median (IQR)	*p*	*p_T1-T2_*	*p_T1-T3_*	*p_T2-T3_*
*Unit: portions/day*								
Meat and meat products	1.02 (1.22)	1.28 (0.95)	1.06 (1.15)	0.37 (1.07)	<0.001 ***	0.744	<0.001 ***	0.003 **
Vegetables and salad	2.30 (1.99)	1.96 (1.73)	2.16 (2.14)	2.51 (1.60)	0.135	0.128	0.064	0.722
Fruits	1.17 (1.28)	0.74 (1.20)	1.16 (1.44)	1.45 (1.15)	0.063	0.200	0.018 *	0.336
Wholegrain products	0.25 (0.45)	0.09 (0.49)	0.25 (0.47)	0.33 (0.32)	0.049 *	0.367	0.012 *	0.185
Sweets, salty snacks, sugar-sweetened beverages, alcohol	2.53 (2.24)	2.58 (1.62)	3.12 (2.29)	1.97 (1.83)	0.038 *	0.471	0.030 *	0.032 *
*Unit: grams/day*								
Sodium	1.87 (1.00)	1.91 (0.78)	2.03 (1.39)	1.45 (0.98)	0.017 *	0.529	0.022 *	0.011 *
Dietary fibers	22.20 (17.30)	17.35 (10.48)	19.90 (23.30)	31.00 (17.80)	0.018 *	0.084	0.006 **	0.262
Saturated fatty acids	31.70 (17.10)	34.95 (15.08)	36.30 (17.70)	27.90 (16.23)	0.022 *	0.970	0.011 *	0.028 *
Added sugar	8.01 (7.82)	7.69 (6.81)	9.34 (9.20)	6.49 (7.41)	0.444	0.897	0.304	0.252

${}^{a}$ -FSA-NPS DI: Inverted Food Standards Agency Nutrient Profiling System Dietary Index. We inverted the original FSA-NPS DI scores to make them directly comparable to other food shopping quality indicators. The higher the inverted FSA-NPS DI, the healthier the food shopping. *p* The Kruskal–Wallis test was performed to compare all three tertiles. *p_T1-T2_*, *p_T1-T3_*, *p_T2-T3_* The Mann–Whitney U tests were performed to compare pairwise tertiles. * *p* < 0.05. ** *p* < 0.01. *** *p* < 0.001.

**Table 9 nutrients-14-00159-t009:** Median and interquartile range (IQR) of the *relative* (i.e., per 1000 kcal of) nutritional intake across the tertiles of the inverted Food Standards Agency Nutrient Profiling System Dietary Index (-FSA-NPS DI) ${}^{a}$.

	-FSA-NPS DI tertile							
	Overall (N = 89)	T1 (N = 30)	T2 (N = 29)	T3 (N = 30)				
	Median (IQR)	Median (IQR)	Median (IQR)	Median (IQR)	*p*	*p_T1-T2_*	*p_T1-T3_*	*p_T2-T3_*
*Unit: portions/1000 kcal*								
Meat and meat products	0.57 (0.57)	0.75 (0.35)	0.68 (0.55)	0.31 (0.54)	<0.001 ***	0.611	<0.001 ***	0.002 **
Vegetables and salad	1.19 (1.06)	1.02 (0.96)	1.25 (0.97)	1.50 (1.55)	0.040 *	0.120	0.014 *	0.321
Fruits	0.60 (0.63)	0.49 (0.63)	0.59 (0.59)	0.82 (0.67)	0.008 **	0.190	0.002 **	0.080
Wholegrain products	0.13 (0.22)	0.06 (0.15)	0.15 (0.20)	0.22 (0.24)	0.007 **	0.299	0.002 **	0.043
Sweets, salty snacks, sugar-sweetened beverages, alcohol	1.50 (0.89)	1.59 (0.86)	1.49 (0.88)	1.36 (0.91)	0.165	0.779	0.115	0.097
*Unit: g/1000 kcal*								
Sodium	1.07 (0.36)	1.13(0.25)	1.20 (0.40)	0.93 (0.36)	0.003	0.897	0.003 **	0.004 **
Dietary fibers	13.23 (8.21)	10.52 (4.11)	13.73 (5.43)	18.78 (11.08)	<0.001 ***	0.057	<0.001	0.012 *
Saturated fat	19.54 (6.27)	20.61 (3.68)	19.61 (5.14)	17.01 (5.74)	0.002 **	0.190	<0.001 ***	0.020 **
Added sugar	4.72 (2.64)	4.42 (2.89)	4.72 (3.43)	4.82 (2.86)	0.851	0.767	0.631	0.688

${}^{a}$ -FSA-NPS DI: Inverted Food Standards Agency Nutrient Profiling System Dietary Index. We inverted the original FSA-NPS DI scores to make them directly comparable to other food shopping quality indicators. The higher the inverted FSA-NPS DI, the healthier the food shopping. *p* The Kruskal–Wallis test was performed to compare all three tertiles. *p_T1-T2_*, *p_T1-T3_*, *p_T2-T3_* The Mann–Whitney U tests were performed to compare pairwise tertiles. * *p* < 0.05. ** *p* < 0.01. *** *p* < 0.001.

## Data Availability

The dataset used in this study, including the anonymized FFQ and anonymized digital receipt datasets, can be found here: https://gitlab.ethz.ch/food-coach/shopping-index-comparison, accessed on 15 December 2021.

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
