# Peer review of "Estimating Dietary Intake from Grocery Shopping Data—A Comparative Validation of Relevant Indicators in Switzerland"

_nutrients, 2021, doi:10.3390/nu14010159_

Round 1
Reviewer 1 Report
The manuscript deals with a crucial research topic as dietary assessment is a complex and difficult task to perform. The rationale of validating the food shopping data represents the main strength of the carried-out study and the originality of the paper too. This means that dietary quality has the most important proxy in the food shopping bag composition although it is not the only aspect in dietary assessment (preparation and consumption patterns are also components). So, it is important to validate the tools from the nutritional evaluation perspective. The strength of the present paper relies on this.
The methodology is appropriate.
The only difficulty I have in reviewing the paper is the use of question marks instead of references number and the lack of a list of references. Moreover, tables and figures are not numbered in the text, but there are question marks.
Remarks
The first remarks concern parts that are used in systematic literature reviw.
I would suggest adding “Switzerland” to the title.
Moreover, I would suggest replacing “purchase” with “food shopping”, or simply adding food shopping somewhere as the food shopping bag is the main subject.
Introduction
Lines 43-46. “dietitians and researchers typically rely 44 on traditional, often still paper-based and rather manual dietary assessment approaches. Seven-day weighed food diaries or records and 24-hour recalls are widely used as the primary validation tools of food intake [? ? ? ? ] The high effort involved in collecting and transcribing such manual logs lead to a substantial burden on the users, researchers and health-care practitioners alike. For example, manual diet logging apps (e.g. MyFitnessPal) are only actively used by 8% of smartphone users [? ]. Due to the effort involved, such traditional diet monitoring approaches usually remain short-lived and lead to relatively high costs, high user attrition rates, under-reporting, and self-selection of primarily healthy users, eventually rendering them inappropriate for large-scale epidemiological studies or interventions.”
This part seems to be a little bit outdated as the computer assisted interviews techniques and the self-administered dietary survey tools are quickly increasing, together with the internet of thins tools that record eating behaviors. The EFSA guidance of the EU Menu programme (https://www.efsa.europa.eu/it/efsajournal/pub/3944) too, promotes the software for 24h recalls and checking paper food diaries. Please, consider further considerations as these tools have pros and cons but are increasingly used and several examples in literature are available. This is outdated also considering that in the methods a web mediated tool is reported.
Lines 54-71. I am not able to check the literature cited, but the FFQ use is to be evaluated according to the objectives of the study, the current situation of the already collected data, the study framework, and the resources.
Please, check that all these aspects had been considered.
Overall, the past experiences in using Household Budget Surveys data (Trichopoulou A, Androniki A and the DAFNE III GROUP (2003): European Food Availability databank based on household budget surveys The Data Food Networking initiative, European Journal of Public Health, Vol. 13, 3 (Supplement), pp. 24-28) have not been cited but in my opinion is relevant as proposing how to transform budgeted data into dietary variables. This is helpful because purchased foods are not simultaneous consumed at least longer shelf-life products, while short shelf-life products need to be prepared quickly and eventually preserved in the freezer or in oil/vinegar/salt/sugar/other preservatives. So, food preparation routines should be also hypothesized. Finally, this kind of data do not include out-of-home consumption, while out-of-home consumption is recorded in individual dietary survey methods (Orfanos P, Naska A, Rodrigues S, Lopes C, Freisling H, Rohrmann S, Sieri S, Elmadfa I, Lachat C, Gedrich K, Boeing H, Katzke V, Turrini A, Tumino R, Ricceri F, Mattiello A, Palli D, Ocké M, Engeset D, Oltarzewski M, Nilsson LM, Key T, and Trichopoulou A (2016): Eating at restaurants, at work or at home. Is there a difference? A study among adults of eleven European countries in the context of the HECTOR project. European Journal of Clinical Nutrition, Paper #2016EJCN0694R).
Methods
Can the Author elaborate on the sampling method? Did you hypothesize a probabilistic structure? My question borns from the use of the probability measure, so a probabilistic structure should be envisaged.
Did the quality indexes consider the contribution from ultra-processed foods?
The nutri-score is neither referenced nor explained. Can the Author kindly add this information? Moreover, the aggregated nutri-score should be defined and referenced too.
Lines 313-317 “To support the assessment of the suggested purchase indicator, eligible users should have at least a certain minimum of their expected calorific energy intake represented in the purchased groceries within their digital receipts over the course of the four weeks preceding their FFQ.”
How was assessed the minimum for energy? Did the Authors consider that not all foods are eaten in one day, especially long shelf-life foods, and that out-of-home consumption are not surveyed in this approach? Did the Authors consider that non-full food shopping has been done? Were the participants to ask the number of days the food basket was referring to? I understand that the four weeks prior to the FFQ administration were used, but how? Were all food shopping events used? Something else?
Lines 322-323 Did the Authors consider that the participant has exchanged height and weight to obtain such a high value?
Results
Could the Authors elaborate on the following comments?
It is clear that the correlation in general is weak (r <0.5), can you discuss a little bit more this?
The potential discrimination is decidedly higher if based on food groups while nutrients are ubiquitous and then less discriminant. Dietary fiber, the only with r=0.5 among relative values, has a different pattern as it is related to specific food groups. Moreover, the index where nutrients are explicitly included is the only one showing discriminant power for nutrients.
Overall vegetables and fruits (and dietary fiber) discriminate better than others groups or nutrients. This can be important on the public health perspective and the easiness to monitor purchase and dietary quality. In fact, both vegetable consumption and fruit consumption were included in the ECHIM list of indicators as n.49 and 50. (https://ec.europa.eu/health/sites/default/files/indicators/docs/echi_shortlist_by_policy_area_en.pdf).
Table 8 and 9 are not commented at all.
Discussion
Lines 540-544 “As the food categorization frameworks between studies might differ and were defined on other regions than Switzerland, this effect might have caused an underestimation of added sugar in the context of this study. Coherent category definitions seem pivotal as we found that the performance results of the GPQI and HPI indicators are very sensitive to changes in the definition of refined grains for example.”
Food grouping is crucial and then using a shared food coding system is extremely helpful. So some references can be added, especially The food classification and description system FoodEx 2 (revision 2) https://efsa.onlinelibrary.wiley.com/doi/abs/10.2903/sp.efsa.2015.EN-804
Lines 549-553 “To ensure the reproducability of future work, authors should publish their food category definitions as detailed information as possible. Machine learning based tools could for example support the (semi-)automatic correction of errors in food composition databases and support the correct identification of products present in digital receipts.”
Please, add references, as tools of such a kind have already been developed.
Minor remark
A typos in line 549 “reproducability” should be changed into “reproducibility”
Author Response
Reviewer 1:
R1.1 The manuscript deals with a crucial research topic as dietary assessment is a complex and difficult task to perform. The rationale of validating the food shopping data represents the main strength of the carried-out study and the originality of the paper too. This means that dietary quality has the most important proxy in the food shopping bag composition although it is not the only aspect in dietary assessment (preparation and consumption patterns are also components). So, it is important to validate the tools from the nutritional evaluation perspective. The strength of the present paper relies on this. The methodology is appropriate.
We thank the reviewer for the comment and agree with the reviewer about the importance and contribution of this paper to the field for food-relevant research.
R1.2 The only difficulty I have in reviewing the paper is the use of question marks instead of references number and the lack of a list of references. Moreover, tables and figures are not numbered in the text, but there are question marks.
We thank the reviewer for the comment. The [?] were caused by ill-formated citations. We double checked the references and they should be in order now. In addition, the tables and figures are now numbered in the text. The list of references is at the end of the paper.
R1.3 I would suggest adding “Switzerland” to the title.
We thank the reviewer for the comment. We have changed the title to ‘Estimating Dietary Intake from Grocery Purchase Data - A Comparative Validation of Relevant Indicators in Switzerland.’
R1.4 Moreover, I would suggest replacing “purchase” with “food shopping”, or simply adding food shopping somewhere as the food shopping bag is the main subject.
We thank the reviewer for the comment. We changed the ‘purchase’ to ‘food shopping’ in the entire manuscript to make it clear and consistent.
R1.5 Lines 43-46. “dietitians and researchers typically rely on traditional, often still paper-based and rather manual dietary assessment approaches. Seven-day weighed food diaries or records and 24-hour recalls are widely used as the primary validation tools of food intake [? ? ? ? ] The high effort involved in collecting and transcribing such manual logs lead to a substantial burden on the users, researchers and health-care practitioners alike. For example, manual diet logging apps (e.g. MyFitnessPal) are only actively used by 8% of smartphone users [? ]. Due to the effort involved, such traditional diet monitoring approaches usually remain short-lived and lead to relatively high costs, high user attrition rates, under-reporting, and self-selection of primarily healthy users, eventually rendering them inappropriate for large-scale epidemiological studies or interventions.” This part seems to be a little bit outdated as the computer assisted interviews techniques and the self-administered dietary survey tools are quickly increasing, together with the internet of things tools that record eating behaviors. The EFSA guidance of the EU Menu programme (https://www.efsa.europa.eu/it/efsajournal/pub/3944) too, promotes the software for 24h recalls and checking paper food diaries. Please, consider further considerations as these tools have pros and cons but are increasingly used and several examples in literature are available. This is outdated also considering that in the methods a web mediated tool is reported.
We thank the reviewer for the comment. First, we changed the ‘traditional, still paper-based and rather manual dietary assessment approaches’ to ‘self-reported and laborious dietary assessment’. Second, we put the main focus on the disadvantages of the manual efforts involved in self-tracking, e.g. high user attrition, recall bias and poor sustainability. Hence, we believe that the assessment of automatically captured digital receipts can be seen as a scalable alternative to monitoring food choices. Although the accuracy of assessing digital receipts might not be comparable to analysing bio-samples (e.g. blood or spot urine) or even 24h food diaries, the novel approach is more convenient, allows capturing of a long and consistent history (up to 2-year history + new receipts), does not suffer from recall bias and allows the assessment of multi-dimensional data such as timestamps and prices that can support the design of relevant interventions. Therefore, the assessment of digital receipts can be seen as a promising scalable alternative to manual food logging in monitoring food choices.
R1.6 Lines 54-71. I am not able to check the literature cited, but the FFQ used is to be evaluated according to the objectives of the study, the current situation of the already collected data, the study framework, and the resources. Please, check that all these aspects had been considered.
We thank the reviewer for the comment. The FFQ used in the study to validate the digital receipts can be found here ( Link: https://gitlab.ethz.ch/jingwu/shopping-index-comparison/-/blob/master/User_Survey_and_Food_Frequency_Questionnaire__1_.pdf). The FFQ instrument has been validated prior to our study and the validation study also took place in Switzerland (see Steinemann et. al. (2017): Relative validation of a food frequency questionnaire to estimate food intake in an adult population). The validation of the FFQ allows for inferring that it correlates with actual dietary intake. Thus, demonstrating the correlation between digital receipts and the FFQ would allow for inferring that digital receipts correlate with actual food intake. The study data, its framework and resources are in line with other validation studies in the field (i.e. HETI, GPQI, HPI). In addition, Prof. Christine Brombach who also supervised the FFQ validation study has served as a co-author in the study, ensuring a high validity of the conducted analyses.
R1.7 Overall, the past experiences in using Household Budget Surveys data (Trichopoulou A, Androniki A and the DAFNE III GROUP (2003): European Food Availability databank based on household budget surveys The Data Food Networking initiative, European Journal of Public Health, Vol. 13, 3 (Supplement), pp. 24-28) have not been cited but in my opinion is relevant as proposing how to transform budgeted data into dietary variables. This is helpful because purchased foods are not simultaneous consumed at least longer shelf-life products, while short shelf-life products need to be prepared quickly and eventually preserved in the freezer or in oil/vinegar/salt/sugar/other preservatives. So, food preparation routines should be also hypothesized. Finally, this kind of data do not include out-of-home consumption, while out-of-home consumption is recorded in individual dietary survey methods (Orfanos P, Naska A, Rodrigues S, Lopes C, Freisling H, Rohrmann S, Sieri S, Elmadfa I, Lachat C, Gedrich K, Boeing H, Katzke V, Turrini A, Tumino R, Ricceri F, Mattiello A, Palli D, Ocké M, Engeset D, Oltarzewski M, Nilsson LM, Key T, and Trichopoulou A (2016): Eating at restaurants, at work or at home. Is there a difference? A study among adults of eleven European countries in the context of the HECTOR project. European Journal of Clinical Nutrition, Paper #2016EJCN0694R).
We thank the reviewer for the comments. The goal of using digital receipts is to identify healthy/unhealthy patterns in people’s food choices, but not to estimate the real food intake. The assessment of digital receipts as a proxy for individual food choices is based on partial data, as out of home consumption, delayed consumption, food waste, or food preparation methods are not inherently captured by purchase data. As it has been shown in previous studies (e.g. HETI, GPQI, HPI, HEI-2010, see more in the manuscript), applying statistical methods on partial data allows the inference of relative distributions (e.g. expenditure share, (food or nutrients)/1000 kcal) that correlated well with actual food intake. In essence, this approach is similar to assessing physical activity via pedometers (i.e. step counting). Pedometers also do not capture 100% of a user’s physical activity as pedometers do not capture bicycle rides or swimming. Still, the individual daily step rate correlates well with a patient’s overall physical activity level and hence is used in many interventions. The relative distribution of the food choices made at the supermarket shelves typically stand for the vast majority of food choices (especially in Switzerland, where ca. 80% of sodium intake originates from packaged food products sourced at supermarkets), and can therefore be used as an acceptable, scalable proxy for food choices. We are still evaluating statistical approaches to reflect gender, food waste behavior and out-of-home behavior into the model to further increase the accuracy of the suggested method, but will present these results in future studies.
R1.8 Can the Author elaborate on the sampling method? Did you hypothesize a probabilistic structure? My question borns from the use of the probability measure, so a probabilistic structure should be envisaged.
We thank the reviewer for the comment. We describe the participant recruiting methods in subsection ‘digital receipt integration’ and ‘study participants’, please also see Figure 1 and Figure 2. We publicly recruited volunteers who could donate their digital receipt data and complete a web-mediated FFQ. As a token of gratitude, all participants were rewarded with a financial compensation of 20 CHF (Swiss Francs). The call to join the study was advertised online and offline. We conducted eligibility criteria on all collected FFQ and digital receipt data. As such, the sampling approach is not based on a probabilistic structure. As it can be seen in Table 2, 76.4% of all volunteers were male and hence we do not claim to have recruited a representative sample of the Swiss population. Still, we believe the results of the study confirm that digital receipt based food choice monitoring can be considered a promising alternative or complementation to contemporary food choice monitoring based on 24h diaries or bio-sampling. In the future and with a larger sample, we could use stratified sampling to further validate the potential of digital receipt based food choice monitoring across all socio-demographic segments separately.
R1.9 Did the quality indexes consider the contribution from ultra-processed foods?
We thank the reviewer for the comment. All indices have some consideration of ultra-processed foods. Each of the five scoring indices (FSA-NPS DI, GPQI, HEI, HTI, HPI) has a unique composition. Since ultra-processed foods are captured well by the digital receipts (e.g. ice-cream, ready-to-eat meals, chocolate bars, processed sausages, etc.), all of the indices consider ultra-processed foods to some extent. We followed the original definition of all given shopping indicators. The detailed food involved in each quality index can be found here: https://gitlab.ethz.ch/jingwu/shopping-index-comparison/-/blob/master/Shopping_indicator_scoring_systems.xlsx. We posted the frameworks publicly on Gitlab and attached the link to the manuscript as well. For example, the FSA-NPS DI considers the density of sugar, calories, sodium and saturated fats within consumed products. As ultra-processed foods usually score high in those dimensions, they are well captured by the FSA-NPS DI. In contrast, the other indices capture ultra processed foods via the consumed products category affiliation: e.g. the GPQI considered Processed meats and Sweets and Sodas separately.
R1.10 The nutri-score is neither referenced nor explained. Can the Author kindly add this information? Moreover, the aggregated nutri-score should be defined and referenced too.
We thank the author for the comment.
The Nutri-Score [Chantal, J., Hercberg, S., & World Health Organization. (2017). Development of a new front-of-pack nutrition label in France: the five-colour Nutri-Score. Public Health Panorama, 3(04), 712-725.] framework is a well-adopted measure for the nutritional quality of packaged food products. It scores food products based on their respective prevalence of healthy (i.e. protein, dietary fiber, fruit & vegetables content ) and unhealthy nutrients (i.e. sodium, saturated fats, sugars) as well as their caloric density (i.e. kcal per 100g of the product). The front-of-package label is color-coded based on the FSA-NPS score. The color and the letter A-E are for communicating the healthiness of the products with consumers in an easy-to-understand manner. To extend the nutri-score framework to unpackaged foods, we can use the FSA-NPS DI, as defined in Validation of the FSA nutrient profiling system dietary index in French adults—findings from SUVIMAX study.
For displaying and faster processing for the user experience on the BAM website, we simplified the Nutri-Score framework by using a weight-based 5-letter system ( A=0.5, B=1.5, C=2.5, D=3.5, E=4.5). For each basket, the weight-weighted average of all products was calculated and displayed on a chart, as shown in Figure 3. A careful review demonstrated that the original framework and the simplified framework yielded very similar results. The simplified version was chosen for simplicity.
We integrated these clarifications into the manuscript.
R1.11 Lines 313-317 “To support the assessment of the suggested purchase indicator, eligible users should have at least a certain minimum of their expected calorific energy intake represented in the purchased groceries within their digital receipts over the course of the four weeks preceding their FFQ.”
How was assessed the minimum for energy? Did the Authors consider that not all foods are eaten in one day, especially long shelf-life foods, and that out-of-home consumption are not surveyed in this approach? Did the Authors consider that non-full food shopping has been done? Were the participants to ask the number of days the food basket was referring to? I understand that the four weeks prior to the FFQ administration were used, but how? Were all food shopping events used? Something else?
We thank the author for the comment. First we would like to explain the data collection, before explaining the minimum amount of energy used to identify eligible users.
All digital receipts were imported automatically after participants joined the project. This means, no manual receipt collection or transcription was conducted. Each digital receipt contains the corresponding date of the time of purchase. For the purpose of the study, we filtered each user’s purchase history for the four-week time window before joining the study & filling out the FFQ.
Figure 2 shows the inclusion criteria. ‘To ensure that food shopping recorded on loyalty cards is representative of actual food intake, eligible users should shop at least 1/7 of the estimated energy intake of all people who share the loyalty cards with participants, within the four weeks preceding their FFQs. The estimated energy intake of a person who was > 13 years old was 2250 kcal/day, irrespective of gender. That of a child (< 13 years old) was estimated to be 0.575 times the energy requirement of a person who is older than 13, i.e. 2250 kcal/day, irrespective of gender. The four-week time window was in line with that of the FFQ, which was validated to estimate a patient's typical diet on a four-week basis. Consequently, 91 participants were excluded as they did not meet the criteria.’ Please see the citation details in the manuscript.
Indeed, we did not reflect the purchased products’ shelf life or out-of-home consumption in the eligibility criteria explicitly. The 4-week-based analysis might help consider the shelf life factor, because most food products are consumed within 4 weeks after shopping. The minimum threshold for energy in shopped products might help consider the out-of-home consumption to some extent. This is a limitation and we mentioned it in the manuscript’s limitations. This problem is similar in other studies (HETI, HPI, GPQI). We would consider it to improve the assessment of digital receipts in future studies.
As stated in the response to question R1.7, the shopping indices are based on the relative composition of the purchased products. This is again similar to other papers in the field (HETI, HPI, GPQI). We used partial food shopping data on loyalty cards to calculate food shopping indicators, indicating how healthy the food shopping is. The potential of loyalty card data being used in nutrition domain has been mentioned before as well. See Nevalainen, J., Erkkola, M., Saarijärvi, H., Näppilä, T., & Fogelholm, M. (2018). Large-scale loyalty card data in health research. Digital health, 4, 2055207618816898., Jenneson, V., Shute, B., Greenwood, D., Clarke, G., Clark, S., Rains, T., & Morris, M. (2020). Variation in fruit and vegetable purchasing patterns in Leeds: using novel loyalty card transaction data. Proceedings of the Nutrition Society, 79(OCE2).
We integrated these clarifications into the manuscript.
R1.12 Lines 322-323 Did the Authors consider that the participant has exchanged height and weight to obtain such a high value?
We thank the author for the comment. We again checked the values entered by the respective user. The entered values for weight and height were as follows: Height: 88 centimeters , Weight: 85 kg . If the user confused both fields, the corrected BMI would still be: 122 kg/ m2. Therefore, we decided to exclude this participant.
Results. Could the Authors elaborate on the following comments?
R1.13 It is clear that the correlation in general is weak (r <0.5), can you discuss a little bit more this?
We thank the reviewer for the comment. In the nutrition domain, a pearson’s r of 0.1-0.3 is considered to be weak and that of 0.3-0.5 is considered to be moderate. To be more accurate, we changed our wording in the paper from ‘generally moderately correlated’ to ‘generally from weakly to moderately correlated’.
Following reasons might cause the weak to moderate correlations. First, there is a time lag between eating and food shopping. E.g. a person bought fruits on 31th Jan but ate them over next week. Second, inter-day/week variance is very high and commonly present. For example, a person can eat pizza on a certain day and skip lunch on the next day. A person can also fast for some time for different reasons. Physical activity also influences how much energy a person needs. Third, only 64.05% of receipts were successfully matched. Fourth, the correlations are between raw value of food category intake and aggregated food shopping indicators.
In comparison with bio-sampling (e.g. sodium excretion or blood sampling) and even 24-hour recalls, although digital receipts have a lower accuracy, they are more cost-effective, scalable and suitable for long-term interventions. Thus, they are still a valuable diet monitoring tool.
We integrated the discussion into the manuscript.
R1.14 The potential discrimination is decidedly higher if based on food groups while nutrients are ubiquitous and then less discriminant. Dietary fiber, the only with r=0.5 among relative values, has a different pattern as it is related to specific food groups. Moreover, the index where nutrients are explicitly included is the only one showing discriminant power for nutrients.
Overall vegetables and fruits (and dietary fiber) discriminate better than others groups or nutrients. This can be important on the public health perspective and the easiness to monitor purchase and dietary quality. In fact, both vegetable consumption and fruit consumption were included in the ECHIM list of indicators as n.49 and 50. (https://ec.europa.eu/health/sites/default/files/indicators/docs/echi_shortlist_by_policy_area_en.pdf).
We thank the reviewer for this comment. We agree that the arguments given by reviewer 2 can be plausible explanations for the superior performance of the FSA-NPS DI index. We hence included the following explanation into the discussion section:
The potential discrimination of the observed indices is decidedly higher if an index is leveraging nutrients as well as food groups. It can be seen in Tables 4 and 5 that the FSA-NPS DI and the HEI-2015 both outperform the other indices. Since the compositions of both indices assess nutrients as well as food groups, such a combination seems to be an important prerequisite for discriminating health-relevant food choices within shopping data. When comparing the HEI-2015 and the FSA-NPS DI indices, the selection and weighting of the indices’ structure allows for the interpretation that dietary fiber is an important factor that gives the FSA-NPS DI its superior discriminatory potential. Both indices are the only ones that include both, namely the consumption of fruit and vegetables, into their index calculation. Overall, the results of Tables 4 and 5 suggest that within the observed food groups, the consumption of vegetables and fruits discriminates better than other food groups. Hence assessing consumption of fruit and vegetables might give both indices their higher discriminatory capability. While the HEI-2015 explicitly includes fatty acids, sodium, added sugars and saturated fats, the FSA-NPS DI includes sodium, saturated fats, sugar and dietary fiber. The results of Tables 4 and 5 suggest that within the observed nutrients, the consumption of dietary fiber discriminates better than other nutrients. It can therefor be an explanation that especially its selection of assessing fruit, vegetable and dietary fiber intake (among other nutrients) as well as its weighting mechanism give the FSA-NPS DI its superior discriminatory capability. This interpretation is also supported by the fact that the European Commission recommends its member states the tracking of their consumers’ intake of vegetables and fruits within their European Core Health Indicators (ECHIM) list of health-relevant policy indicators [ECHIM citation]. The use of the FSA-NPS DI, or other similar indicators that allow the non-invasive and scalable monitoring of the dietary quality of purchased food items can hence allow the validation of health-policies as well as be the entry point for continuous just-in-time adaptive dietary behavior change interventions.
R1.15 Table 8 and 9 are not commented at all.
In the manuscript, we are now referencing the Tables 8 and 9 as follows and have integrated it to the manuscript.
To confirm the ability of the FSA-NPS DI to distinguish between users with different dietary habits, the characteristics of the highest, medium and lowest tertile of the FSA-NPS DI were assessed in Table 8 and 9, which shows the absolute and relative i.e. per 1’000 kcal, food intake of the entire sample population and three tertiles as measured by the FFQ. The median and IQRs of were reported because of the non-normal distribution of FSA-NPS DI. The lower the number of FSA-NPS DI, the healthier the baskets should be. T1 has the shopping baskets of the healthiest dietary quality. On both absolute and relative scales, T1 consumed significantly more meat and less dietary fibers compared to T2 and T3.
R1.16 Lines 540-544 “As the food categorization frameworks between studies might differ and were defined on other regions than Switzerland, this effect might have caused an underestimation of added sugar in the context of this study. Coherent category definitions seem pivotal as we found that the performance results of the GPQI and HPI indicators are very sensitive to changes in the definition of refined grains for example.”
Food grouping is crucial and then using a shared food coding system is extremely helpful. So some references can be added, especially The food classification and description system FoodEx 2 (revision 2) https://efsa.onlinelibrary.wiley.com/doi/abs/10.2903/sp.efsa.2015.EN-804.
Lines 549-553 “To ensure the reproducability of future work, authors should publish their food category definitions as detailed information as possible.
We thank the reviewer for this comment. Especially since we were not aware about the FoodEx 2 classification schema, we decided to integrate this aspect into the future work section of the manuscript. We also cited the reference and link, which the reviewer kindly provided.
‘To ensure the reproducibility of similar studies in the future, researchers should adhere to established food classification schemata such as FoodEx2 (revision 2). Albeit FoodEx2 is only defined for the European context, such classification schema can be very helpful in transferring concepts to other regions as well. Ultimately, a global food classification schema would be helpful to reproduce the suggested grocery purchase quality indicators internationally.’
R1.17 Machine learning based tools could for example support the (semi-)automatic correction of errors in food composition databases and support the correct identification of products present in digital receipts. Please, add references, as tools of such a kind have already been developed.
To support the extensive effort required for the correct mapping of digital receipts and food composition databases, machine-learning based algorithms [1] can support the (semi-)automatic correction of errors in food composition databases [2] or the correct identification of products present in digital receipts (e.g. word2vec) [3, 4].
We integrated these references into the manuscript. The references can be found in the manuscript.
R1.18 A typos in line 549 “reproducability” should be changed into “reproducibility”
We thank the reviewer for the comment and this typo has been corrected.
We really appreciate reviewer 1’s thorough review, constructive feedback and helpful recommendations for our manuscript and our future work.

Reviewer 2 Report
This is a very important methods paper needed in the field of food environment research.
Abstract/throughout: "diet-related chronic diseases" is a more accurate term relative to non-communicable. Some of the language around transitions inserts words that are unneeded- e.g. Line 11, "The study hence"; hence is unneeded. Do not start sentences with digits.
Abstract
Line 8: "purchase quality" is not defined and not a known term
Line 18, just state the final sample not the # asked; you could explain the 484 were used to build database
Introduction:
Lines 43-44: "self-reported and laborious" might be more accurate than "traditional" "paper-based"
Line 59: "Similar to weighed food diaries"
Missing in Intro: You do not acknowledge the benefits and short-comings of individual level nutrition data and thus you do not make a case for why household food shopping could be a good measure. You hint at in Line 80 but you do not shape intro around this. From lines 85-105 it seems you are missing acknowledgement that grocery store data represents food to be eaten at home - need to describe how that food influence overall diet.
Lines 130-137: the terms calibration and discrimination seem to replace the terms reliability and validity- is this the case or are they more specific measures of these measure quality indicators? would be helpful if described here.
Materials and Methods:
Line 168: for the 76% of "grocery market" is that in terms of # of stores or sales?
Lines 236-241" The rationale, "to maximize the frequency with which purchased items were identified" needs to be further described in terms of why? That is, why could you not assess all products? Was it because the GTIN to product identification is manual? OR there are just too many products (if so how many?) And when you say you came up with a heuristic, describe explicitly- e.g. how many times did the product need to appear across the data?
Line 241: Why were the 464 used for this process?
Line 246-249 seems like it should be in discussion
Line 261 "ground truth data" is unclear term; perhaps objective measure is better
Page 9: some elements repeat from page 5
Table 1: For sodium, Dietary fiber, Sat Fatty acids and Added sugar provide the amount that represented a "portion" in grams
Line 339: Cite the recommendning agency
Line 347: "realistic" odd word choice. perhaps "representative of typical intake" is what was meant?
Results:
Line 44: Convention is that 0.30 and greater is moderate. Be precise with wording so as to not overstate your results.
Table 4/5: mention that the r values are from Pearson correlation test
Table 6/7 Fairly difficult to interpret. For example unclear in Table 6 what GPQI and Fruits symbol means. Unless another way to display, I think the narrative is enough and would recommend deleting.
Discussion and Conclusion:
489 "ground truth" phrase again
Lines 501-503: Consider wording revision to be less aspirational and more specific with how this can be used.
Line 557-558: odd sentence structure, revise
Line 574: it was either low or moderate correlations (not all moderate)
Author Response
Reviewer 2:
R2.1: This is a very important methods paper needed in the field of food environment research.
We thank the reviewer for the comment and we agree with the reviewer that this paper is important for the methodology in the field of food environment research.
R2.2: Abstract/throughout: "diet-related chronic diseases" is a more accurate term relative to non-communicable. Some of the language around transitions inserts words that are unneeded- e.g. Line 11, "The study hence"; hence is unneeded. Do not start sentences with digits.
We thank the author for the comment. First, we changed the term ‘diet-related non-communicable diseases’ to ‘diet-related chronic diseases in the abstract and throughout the manuscript. In addition, we removed the unnecessary word ‘hence’ from Line 11. Finally, we rephrased the sentence that started with a number.
R2.3: Abstract: Line 8: "purchase quality" is not defined and not a known term
We thank the reviewer for the comment. We agree with the reviewer that the term ‘purchase quality’ can be considered ambiguous. Thus, we rephrase the term ‘purchase quality’ to nutritional quality of food shopping’. This term also considers the feedback from reviewer 1 who refers to ‘food shopping’ rather than ‘grocery purchases’.
In the context of this manuscript, the term nutritional quality of food shopping’ refers to the nutritional quality of food items that consumers bought using their loyalty cards. Although there exists differences in the interpretation of dietary quality of purchased food groups and nutrients, the literature agrees that on average larger quantities of fruits, vegetables and dietary fibers are considered healthy, while energy-dense packaged products high in sodium, (added) sugar and saturated fats are considered unhealthy. The term shall refer to the overall assessment of the dietary quality of the food products recorded on loyalty cards from the study sample within the observation period of this study, i.e. four weeks before finishing the FFQ.
As metrics of quantifying nutritional quality of food shopping, five food shopping indicators are introduced. HEI-2015, HPI, GPQI, HETI are about evaluating the compliance to local dietary guidelines. FSA-NPS DI considers the nutritional content of baskets according to its definition.
We integrated these comments to the manuscript.
R2.4: Line 18, just state the final sample not the # asked; you could explain the 484 were used to build database
We thank the reviewer for the comment. We decided to only keep the number of eligible participants in the abstract. The number of participants who donated their data and contributed to the food composition database is described in the main body only.
R2.5 Introduction: R2.4.1 Lines 43-44: "self-reported and laborious" might be more accurate than "traditional" "paper-based"
We thank the reviewer for the comment and changed the wording accordingly.
R2.6 Line 59: "Similar to weighed food diaries"
We thank the reviewer for the comment and changed the wording accordingly.
R2.7 Missing in Intro: You do not acknowledge the benefits and short-comings of individual level nutrition data and thus you do not make a case for why household food shopping could be a good measure. You hint at in Line 80 but you do not shape intro around this. From lines 85-105 it seems you are missing acknowledgement that grocery store data represents food to be eaten at home - need to describe how that food influence overall diet.
We thank the reviewer for the comment.
We agree that the case for scalable food choice monitoring on household shopping data can be made stronger in the introduction. We hence reorganized the introduction and introduced the benefits and short-comings of individual as well as household level nutrition data that were previously split up into the different subsections (Conventional diet monitoring approaches; Digital receipts). The new introduction can be found in the manuscript.
R2.8 Lines 130-137: the terms calibration and discrimination seem to replace the terms reliability and validity- is this the case or are they more specific measures of these measure quality indicators? would be helpful if described here.
We thank the reviewer for the comment. We defined calibration and discrimination in the section ‘Introduction’ and a more specific one in the section ‘Materials and Methods’.
These two terms are commonly used in the nutrition domain. For instance, in the paper of Nutri-Score (it also mentions FSA-NPS), it has the sentence ‘Moreover, wide variability was observed within food groups; this allows for discrimination of nutritional quality both across groups of foods and within a food group’. Calibration is commonly used in the famous European Prospective Investigation into Cancer and Nutrition (EPIC) studies.
R2.9: Materials and Methods: Line 168: for the 76% of "grocery market" is that in terms of # of stores or sales?
We thank the reviewer for the comment. We double checked the resources and concluded that Switzerland is in a unique position to validate the potential of digital receipt. First, the retail market is dominated by two large supermarket chains, namely Migros and Coop, with a sales share of 70%(see references in the paper). Second, the loyalty cards are used extensively. Around 80% of Coop’s annual sales are achieved with Supercard customers. This has been integrated to the manuscript.
R2.10: Lines 236-241" The rationale, "to maximize the frequency with which purchased items were identified" needs to be further described in terms of why? That is, why could you not assess all products? Was it because the GTIN to product identification is manual? OR there are just too many products (if so how many?) And when you say you came up with a heuristic, describe explicitly- e.g. how many times did the product need to appear across the data? Line 241: Why were the 464 used for this process?
Since both retailers, i.e. Migros and Coop, do not include the GTIN within their digital receipt formats, the setup of the food composition database and the product identification, a heuristic was applied to correctly identify the most frequently purchased food products. In the context of this study, a total of N=464 users were invited to donate their shopping data. This data was used to identify the most frequently occurring food products. In total, 65'391 different products were observed by assessing the entire shopping history of the N=464 users. In total, 5'950 product article identifiers from the digital receipts were mapped to corresponding GTINs. For each of the matched products, its attributes such as nutritional details (e.g. calorific energy and macro-nutrients such as protein, carbohydrate, sugar, fat, saturated fat, dietary fiber, and minerals such as sodium, all per 100 g or ml of product; 1 g corresponds to 1 ml), its logistical data (e.g. product size in g, kg, ml or l), product images, allergens, ingredients were made available for the analysis. These mapped articles correspond to 4'951 of the most frequently occurring products. This is, because all coupons and all non-food items were labelled identically, i.e. the study does not differentiate between different types of coupons or non-food items. These 4'951 labelled product items represent most of the frequently bought food items. Since the two supermarkets also feature non-food items (e.g. plastic bags, napkins, toilet paper etc.) and since some food items cannot be identified (e.g. ‘Menu 1’, ‘Lunch Menu’, etc.), some frequently occurring products are not identified in this study. Still, the overall matching ratio, i.e. the proportion of identified products within the shopping history of the eligible users in the four-week observation period is 69.6%. This shows that in order to capture a majority of the products purchased in digital receipts in Switzerland, less than 10% of the products must be identified in the corresponding food composition database.
We integrated a corresponding comment into the manuscript.
R2.11: Line 246-249 seems like it should be in discussion
We thank the reviewer for the comment and removed the corresponding section there, as requested.
R2.12: Line 261 "ground truth data" is unclear term; perhaps objective measure is better
We thank the reviewer for the comment. All mentions of ‘ground truth’ in the paper have been changed to ‘objective measurement’.
R2.13: Page 9: some elements repeat from page 5
We thank the reviewer for the comment. The repeated part containing the study protocol and participant description was removed from page 9.
R2.14 Table 1: For sodium, Dietary fiber, Sat Fatty acids and Added sugar provide the amount that represented a "portion" in grams
We thank the reviewer for the comment. Unfortunately, the corresponding column in Table 1 contained a labelling error. The mistake has been corrected in the revised version of the manuscript. The correct term is grams/day for the referenced nutrients. Portions were only used for food groups and introduced correctly in the manuscript.
R.2.15: Line 339: Cite the recommendation agency
We thank the reviewer for the comment. This was recommended by the Swiss Food Pyramid, suggested by the Swiss Society for Nutrition.
We have integrated the respective citation into the manuscript.
R.2.16: Line 347: "realistic" odd word choice. perhaps "representative of typical intake" is what was meant?
We thank the reviewer for the comment and changed the wording.
R2.17: Results: Line 44: Convention is that 0.30 and greater is moderate. Be precise with wording so as to not overstate your results.
We thank the reviewer for the comment. We agree that it is important to reflect the results objectively. Since only some Pearson correlations are above 0.3, we decided to change the corresponding interpretation of the values to ‘weakly to moderately correlated’, rather than ‘moderately correlated’. We observed the strongest correlations between HEI-2015 and dietary fiber, when using absolute dietary intake, and that between FSA-NPS DI and dietary fiber, when using relative dietary intake.
R2.18: Table 4/5: mention that the r values are from Pearson correlation test
We thank the reviewer for the comment. We agree with the reviewer that it is important to be explicit about the statistical methods that were used in the context of the study. We corrected the term ‘correlation’ in Tables 4 and 5 to ‘​​Pearson correlation coefficients’ to make the context explicitly clear.
R2.19 Table 6/7 Fairly difficult to interpret. For example unclear in Table 6 what GPQI and Fruits symbol means. Unless another way to display, I think the narrative is enough and would recommend deleting.
We thank the reviewer for the comment. We agree that the Tables 6 and 7 needed editing to make them self-explanatory. We therefore decided to add a brief introduction of the food shopping indicators and a definition of different food groups in the footnotes.
R2.20: Discussion and Conclusion: R2.8.1 489 "ground truth" phrase again
We thank the reviewer for the comment. We substituted the term ‘ground truth’ with ‘objective measurement’ throughout the manuscript as described in our answer to R2.13.
R2.21 Lines 501-503: Consider wording revision to be less aspirational and more specific with how this can be used.
We thank the reviewer for the comment.
We rephrased the corresponding sentence to: ‘ Second, the consistent out-performance of the FSA-NPS DI in nutritional intake calibration and discrimination demonstrates that it could be a good choice for general purpose of use, particularly when developing tools for long-term diet monitoring and intervention. ’
R2.22: Line 557-558: odd sentence structure, revise
We thank the reviewer for the comment. We rephrased the corresponding sentence in the manuscript.
Old version:
Data portability as mandated by the GDPR [39] itself is not sufficient, but if products’ GTINs were present in digital receipts, novel post-purchase services could help consumers around the world make healthier and potentially more sustainable food choices.
New version:
This study also calls for the integration of product identifiers into digital receipt standards. Unfortunately, retailers are currently not required to integrate relevant product identifiers such as the GTIN into their digital receipt structures. Therefore, the identification of text-based product identifiers within food product composition databases requires lots of manual effort. It would therefore be beneficial for the development of scalable digital receipt-based food choice monitoring and interventions if regulators would extend the right for data portability as mandated by the GDPR [GDPR2018] and mandate the use of product identifiers within digital receipt standards.
R2.23: Line 574: it was either low or moderate correlations (not all moderate)
We thank the reviewer for the comment and changed ‘moderately’ to ‘weakly to moderately’. This is similar to R2.17.
We integrated these lines of argumentation into the discussion section of the manuscript.

This manuscript is a resubmission of an earlier submission. The following is a list of the peer review reports and author responses from that submission.
Round 1
Reviewer 1 Report
Overall, the topic of this manuscript was interesting and relevant to the field. However, my concerns with the quality of the writing, unclear research questions and gaps, confusingly written methods, and the absence of standalone tables/figures made it very difficult to follow the main objectives of the manuscript. I believe it could be improved with significant editing for content and writing style and be of interest to the field.
The introduction is very long, brings up many themes without clearly relating them, and fails to discuss a research gap, research question, and hypotheses. References are missing throughout. The methods does not clearly define calibration capacity or discrimination potential nor the various diet quality indices clearly. Additionally things that don't belong in the methods are there (e.g. lines 244-248). It is unclear if this is a repurposed dissertation chapter based on the reference to the previous chapter in line 314. Some of the results seem questionable- it seems implausible in table 1 for participants to have 2.9 servings/d of sweets yet only 10 g added sugar. Other values also seem low. In general, tables and figures aren't standalone and there are an excessive number. Given the weaknesses in writing and flow, the overall message and significance of the manuscript, which seem like an interesting topic was very difficult to follow.
Reviewer 2 Report
This is a novel study that provides a creative example of the use of electronic purchase records. The authors have found that FSA-NPS DI showed to be the best estimator of the purchase quality indicators when using the electronic purchase records.
My comments are as follows:
- Please correct references, table numbers and figure numbers where they show as question marks. Blank table and figure numbers made it very difficult to follow.
- In the Introduction, the authors mentioned " as described in 2.1.1", which could not be found. Please correct this.
- In all figures, please spell out the abbreviations in the footnote.
- The description of the discrimination capacity method is confusing. How was compliance measured? Have the authors considered using the area under the curve to assess the discrimination capacity? Why was the significance of Kruskal-Wallis test included in the score if the significance of the pairwise comparisons were taken into account? Would this not be double counting?